# Contact-engineered reconfigurable two-dimensional Schottky junction field-effect transistor with low leakage currents

Yaoqiang Zhou[1], Lei Tong[1], Zefeng Chen [2], Li Tao[3], Yue Pang[1] & Jian-Bin Xu [1] ✉

Two-dimensional (2D) materials have been considered promising candidates for future low power-dissipation and reconfigurable integrated circuit applications. However, 2D transistors with intrinsic ambipolar transport polarity are usually affected by large off-state leakage currents and small on/off ratios. Here, we report the realization of a reconfigurable Schottky junction field-effect transistor (SJFET) in an asymmetric van der Waals contact geometry, showing a balanced and switchable n- and p-unipolarity with the $I_{ds}$ on/off ratio kept >$10^6$. Meanwhile, the static leakage power consumption was suppressed to $10^{-5}$ nW. The SJFET worked as a reversible Schottky rectifier with an ideality factor of ~1.0 and a tuned rectifying ratio from $3 \times 10^6$ to $2.5 \times 10^{-6}$. This empowered the SJFET with a reconfigurable photovoltaic performance in which the sign of the open-circuit voltage and photo-responsivity were substantially switched. This polarity-reversible SJFET paves an alternative way to develop reconfigurable 2D devices for low-power-consumption photovoltaic logic circuits.

Emerging reconfigurable technologies based on two-dimensional (2D) materials are expected to empower next-generation electronic devices with lower power consumption and higher expressive capability[1-5]. Different from conventional complementary metal-oxide semiconductor (CMOS) technologies, 2D-materials-based transistors show the dopant-free polarity control that the conduction mechanism reversibly switches between n-type and p-type operation modes under gate-voltage sweeps[6-8]. These transistors are regarded as Schottky junction field-effect transistors (SJFETs), in which the carrier injection is determined by the Schottky barrier (SB) formed at the metal/semiconductor interfaces (MSIs)[9,10]. However, this ambipolar behavior renders the SJFET hard to be switched off properly[11]. Especially in downscaling of 2D SJFET, the strong electrostatic control can shrink the off-state SB, resulting in a larger leakage current and a smaller current on/off ratio, which hampers its application towards the large-scale integration and low static power consumption[12,13].

Contact engineering as an effective modulation method has been proposed to suppress ambipolarity under gate-voltage sweeps and achieve an ultra-low off-state current in the SJFET[14-17]. The ambipolar-to-unipolar conversion of transport polarity control in 2D SJFET has been realized by introducing dual gates under the source and drain, respectively, where the injection and conduction of carriers can be individually modulated[4,18]. But the multi-gate layout in integrated circuits impeded device scaling due to gate-length limitations. The insertion of an atomically thin insulator layer such as hexagonal boron nitride or the air gap in MSI was also effective in improving the interface quality and suppressing the drain leakage[19,20]. The tunneling conduction mechanism, however, inevitably decreases the on-state current density. An easier method to build unipolar SJFET with an uncompromised on-/off-state current ratio is to use asymmetric source/drain electrodes with different work functions[21,22]. However, the strong Fermi-level pinning induced by the metal-induced gap states (MIGs) and trap-induced gap states between deposited metal/

[1]Department of Electronic Engineering and Materials Science and Technology Research Center, The Chinese University of Hong Kong, Hong Kong, SAR, China. [2]School of Optoelectronic Science and Engineering and Collaborative Innovation Center of Suzhou Nano Science and Technology, Soochow University, 215006 Suzhou, China. [3]Key Lab of Advanced Optoelectronic Quantum Architecture and Measurement (Ministry of Education), School of Physics, Beijing Institute of Technology, 100081 Beijing, China. ✉e-mail: jbxu@ee.cuhk.edu.hk

2D semiconductor interfaces usually generates an unpredictable SB height and brings the uncertainty to control the unipolarity of SJFETs[21].

Emerging layered semimetals/metals provide a state-of-the-art approach for building the MSI to achieve ambipolar-to-unipolar conversion in van der Waals (vdWs) layered SJFETs. The bond-free integration is unaffected by lattice mismatch, or defects induced in metal deposition process to avoid trap states in the MSI[7], making the SB height more controllable. Furthermore, semimetals with a near-zero density of states at the Fermi level have been verified to avoid MIGs and achieve the ideal MSIs[23]. Graphene as a typical Dirac semimetal, whose Fermi energy can be effectively tuned by electrostatic gating has the potential to build the reconfigurable barrier transistor[24]. Beyond graphene, the family of transition metal dichalcogenide (TMD) also furnishes a rich variety of semimetals, e.g., 1T'-MoTe$_2$[25,26], 1T'-WTe$_2$[27,28], 1T'-PtSe$_2$[29], and 2H-NbSe$_2$[30]. These semimetals possess a broad range of work functions and are expected to create predictable and high-quality all vdWs Schottky junctions[31,32].

Here, we reported on a runtime reconfigurable WSe$_2$ SJFET with epitaxially-grown WTe$_2$ and mechanically exfoliated multi-layer graphene (MGr) contacts, in which the WTe$_2$ contact effectively suppressed the carrier injection to realize the ambipolar-to-unipolar polarity conversion controlled by the single bottom gate. Because carrier injection only allowed tunneling from the MGr contact, the WSe$_2$ SJFET shows an alternative carrier polarity between n-type and p-type, by applying positive and negative source-drain voltage $V_{ds}$, respectively. The SJFET under both p-type and n-type unipolarity conditions suppressed the leakage currents to $2 \times 10^{-10}$ μA/μm, while the controllable $I_{ds}$ on/off ratios with a maximum of $10^6$ retained. The static power consumption induced by the leakage off-state $I_{ds}$ current was suppressed to $10^{-5}$ nW. The SJFET also worked as an electrically gate-tunable Schottky rectifier with a near-unity ideality factor of ~1.0 and a high rectifying ratio of $3\times10^6$. The asymmetrically contacted

SJFET showed a reconfigurable photovoltaic performance with the open-circuit voltage $V_{oc}$ substantially tuned from 0.29 V to −0.47 V and the self-powered photoresponsivity markedly tuned from 61.7 to −12.7 mA/W. Both the negatively and positively gate-biased asymmetric photodiodes showed high filling factors with a maximum of 0.68, indicating larger shunt resistance and smaller leakage. As a facile design method, the WTe$_2$/MGr contact strategy is also applicable to other 2D materials such as WTe$_2$/MoS$_2$ gate-tunable n-type Schottky diode, to boost 2D reconfigurable SJFETs in applications towards low-static-power-consumption and run-time reversible photovoltaic electronics.

## Results and discusssion
### The conversion from ambipolarity to reconfigurable unipolarity
The ambipolarity of the SJFET is attributed to the energy level alignments and evolution at both the source and drain Schottky contacts. Figure 1a shows the schematic of the SJFET with intrinsic ambipolar transport polarity, with its band alignment evolution and the corresponding ambipolar transfer curves schematically shown in Fig. 1b. According to the general theory of the Schottky barrier based on 2D semiconductors[9], there are two main types of carrier injection mechanisms: (1) thermionic emission (TE) when the gate voltage $V_g$ is smaller than flat band voltage $V_{FB}$, which is given by Eqs. (1) and (2):

$$I_{thermal} \approx AT^2 \exp\left(\frac{q\varphi_B}{k_B T}\right) = AT^2 \exp\left(\frac{q(\varphi_{SB-n} + \psi_S)}{k_B T}\right) \quad (1)$$

$$q\psi_S \approx \left|\frac{V_g - V_{FB}}{\gamma}\right|, \gamma \approx 1 + \frac{C_s + C_{it}}{C_{ox}} \quad (2)$$

where is the $\varphi_B$ barrier height, $\psi_s$ is the surface potential, $\gamma$ is the inverse band movement factor and calculated by the semiconductor capacitance $C_S$, the interface trap capacitance $C_{it}$, and the oxide

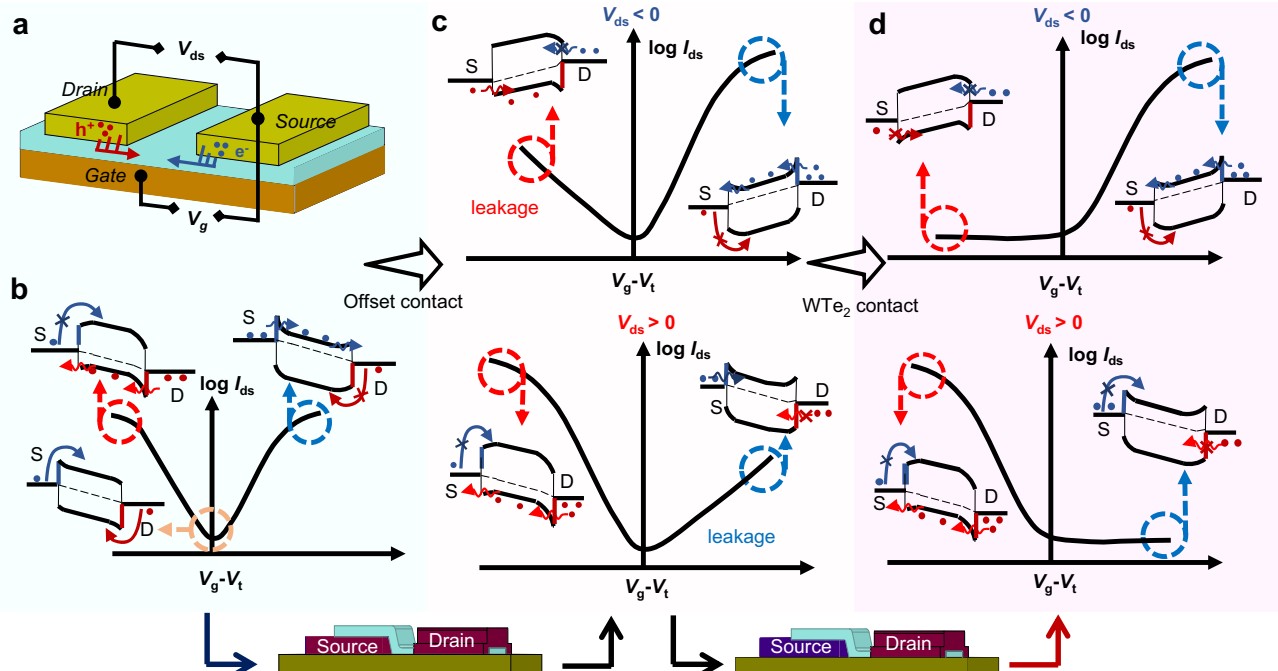

**Fig. 1 | The transition from ambipolarity to unipolarity behavior of the Schottky junction field-effect transistor (SJFET). a** Schematic of global gated SJFET with symmetric drain (D) and source (S) contacts. **b** Schematic of ambipolar transport behavior and corresponding band structure diagram. Both n- and p-branch on-state currents are attributed to the tunneling current due to the barrier reduction and thinning. TE represents thermionic emission. DT represents directing tunneling. **c** Offset contact geometry enabling p-type and n-type transport behavior and the corresponding band structure diagram. **d** Suppression of leakage currents by utilizing WTe$_2$ bottom contacts. The red and blue circles represent the hole and electron, respectively. The arrows represent the carrier injection. $V_g$ and $V_{ds}$ are the gate voltage and drain–source voltage, respectively. $I_{ds}$ is the drain–source current. $V_t$ is the threshold voltage.

capacitance $C_{ox}$. (2) thermally assisted Fowler–Nordheim tunneling (FNT) and direct tunneling (DT) when the $V_g$ exceeds the $V_{FB}$, which is given by Eqs. (3) and (4):

$$I_{tunnel} = \frac{2h}{q} \int_{q\varphi_s}^{\varphi_{SB-n}} f(E) M_{2D}(E) T_{WBK}(E) dE \qquad (3)$$

$$T_{WBK}(E) = \exp\left( -\frac{8\pi}{3h} \sqrt{2m^*(\varphi_{SB-n}-E)^3} \frac{\lambda_{SB}}{q\psi_S} \right) \qquad (4)$$

where is the $f(E)$ is the Fermi–Dirac distribution of the contact metal, $M_{2D}(E)$ is the number of 2D conduction modes in the semiconducting channel, and the $T_{WKB}(E)$ is the SB transmission probability, and the $\lambda_{SB}$ is the characteristic length. As shown in Fig. 1b, when a positive $V_g > V_{FB}$ is applied, the $\psi_s$ increases and the energy-band diagram is bent upwards to enable an electron tunneling current $I_{tunnel}$ from source to drain, which significantly contributes to the on-state current $I_{on}$ on the n-branch. Reversely, when a negative $V_g < V_{FB}$ is applied, the $\psi_s$ decreases, and the downward-bent band induced hole injection from the drain through direct tunneling, which contributes to the $I_{on}$ on the p-branch. By setting $V_g$ at zero, the DT current is suppressed, and only the lower TE current contributes to the off-state current due to the larger width of the contact barrier. Hence, when $V_g$ exceeds the $V_{FB}$, both n- and p-$I_{on}$ are dominated by the $I_{tunnel}$ through the narrowed barrier width.

To achieve the ambipolar-to-unipolar transition, we proposed an offset contact strategy to suppress the carrier injection from the source by utilizing the shielding effect of the bottom electrode to avoid electrostatically $\psi_s$ shifting (Fig. 1c). Firstly, we designed an SJFET with the MGr offset contacts, as shown in Supplementary Fig. 1a, in which the source and drain were contacted at the bottom and the top surfaces of the WSe$_2$ channel, respectively. Supplementary Fig. 1b–c show the transfer curves of the WSe$_2$ FET at various $V_{ds}$ with the offset contacts. When $V_{ds} < 0$, the WSe$_2$ transistor showed a p-type transport polarity with an on-state $I_{ds}$ of 0.18 µA/µm. Reversely, the WSe$_2$ transistor showed an n-type polarity when $V_{ds} > 0$. The output curves at negative and positive $V_g$ also indicated that the back-to-back junctions were asymmetrically modulated, as shown in Supplementary Fig. 1d. However, the carrier injection from the source side was only suppressed slightly, as shown in Supplementary Fig. 1e–g, which induced a higher off-state $I_{ds}$ of $0.4 \times 10^{-2}$ µA/µm and a lower on/off ratio (~10) for the MGr-contacted WSe$_2$ SJFET. This weak ambipolar-to-unipolar transition was hard to be simply explained by the air-gap-induced barrier widening[20]. The contact interaction between the WSe$_2$ and the bottom contact also played an important role to repress the leakage current, which will be discussed in Part III. According to our previous report based on the MGr/MoS$_2$/WTe$_2$ vertical junction[33], WTe$_2$ possessing an appropriate work function and weak interlayer interaction with WSe$_2$ is expected to enlarge the tunneling width of SB at WTe$_2$/WSe$_2$ interface. Therefore, we can optimize the bottom contact material by utilizing WTe$_2$ to suppress the leakage currents and fulfill the ambipolar-to-unipolar transition by only allowing carrier injection from the top contact side, as shown in Fig. 1d.

**Preparation and electrical characterization of WTe$_2$ contacts**
To obtain a high-quality WTe$_2$ bottom electrode, we proposed a one-step epitaxial growth method to prepare large-size WTe$_2$. Figure 2a shows the photograph of epitaxially-grown WTe$_2$ in which thicker MoTe$_2$ flakes were synthesized first and worked as growth seeds to provide nucleation sites and reduce the nucleation energy barrier of WTe$_2$. Compared to direct WTe$_2$ growth, the epitaxial growth of WTe$_2$ showed a large-size and high-quality surface and clear W(Mo)Te$_2$ interface, advantageously reducing the unexpected Fermi level pinning and controlling the contact barrier precisely. More information about the sample growth and characterization is in the Methods

section and Supplementary Note 1. Figure 2b–d show the Raman intensity mapping indicating the heterostructure properties and excellent uniformity of MoTe$_2$ and epitaxial WTe$_2$. The surface quality of the WTe$_2$ was probed by atomic force microscope (AFM), as shown in Fig. 2e. Both the thin WTe$_2$ at the edge and the thick MoTe$_2$ possessed smooth surfaces, which were expected to work as the vdWs Schottky contact with a clean and desired interface.

As a promising candidate to build the semimetal-semiconductor junction, the potential and resistance of epitaxially-grown WTe$_2$ were further investigated. Figure 2f shows the surface-potential image of WTe$_2$ measured by Kelvin probe force microscopy (KPFM). The edge WTe$_2$ possessed a higher surface potential compared to the thick MoTe$_2$ region and the difference in potential between MoTe$_2$ and WTe$_2$ was 79 meV. The work function (WF) of WTe$_2$ was -5.17 eV by using Au film (WF$_{Au}$ = 5.1 eV) as the reference (Fig. 2g). Figure 2h and i show the $I_d$–$V_d$ curves of W(Mo)Te$_2$ with various thicknesses characterized by 2-terminal and 4-terminal methods, respectively. The measurements are detailed in Supplementary Fig. 5. Since the 4-terminal resistance ($R_{4T}$) eliminated the contact resistance, it was used to assess the intrinsic electrical resistance of the WTe$_2$ (The details were discussed in Supplementary Note 2). The WTe$_2$ with a thickness $t$ of ~5 nm showed the highest resistance of about $R_{4T}$ = 9.41 kΩ. For the sample with thickness between $t = 5$–20 nm, the resistance decreased to 1.95 kΩ and the bulk sample ($t > 20$ nm) possessed the lowest conductivity of 0.58 kΩ (Fig. 2i). Compared to the $R_{4T}$, the 2-terminal resistance ($R_{2T}$) showed a more pronounced change with thickness, indicating the contact resistance $2R_{contact} = R_{2T} - R_{4T}$ between the transferred Au film and W(Mo)Te$_2$ increased with the decreased thickness, as shown in the inset of Fig. 2j.

Further, the current density ($V_{ds} = 0.1$ V) of WTe$_2$ devices at different temperatures was measured by the 2-terminal method (Supplementary Figs. 6–7). The current density of WTe$_2$ with a small thickness of ~5 nm was positively correlated with the temperature. As the thickness increased, the WTe$_2$ exhibited a weak temperature dependence. However, the WTe$_2$ with a thickness larger than 20 nm showed a negative temperature coefficient of current density, as shown in Supplementary Fig. 7b–d. This metal-semiconductor transition was consistent with exfoliated W(Mo)Te$_2$ flakes in previous reports[34], which may be attributed to contact resistance and the surface absorption of the hydroxyl group in ambient conditions. The transfer curves of WTe$_2$ with varied thicknesses also support this transition (Supplementary Fig. 7e–g). The bulk sample exhibited a constant source-drain current $I_{ds}$ when the gate-voltage $V_g$ swept, but the $I_{ds}$ measured in the thin WTe$_2$ were modulated by gate-voltage steadily, showing a weak p-type characteristic, especially in the low-temperature range.

**Carrier injection capability comparison between MGr and WTe$_2$**
To compare the carrier injection capability of different contact materials and geometries, we built the WSe$_2$-SJFET using symmetric top and bottom contacts with the exfoliated MGr and epitaxially grown WTe$_2$. All devices were fabricated by the dry-transfer method to avoid the formation of defect-induced states. Figure 3a, b show the schematics and transfer curves of the top-contacted device at various $V_{ds}$. Compared to the FET with top WTe$_2$ contacts, the FET with MGr top electrodes showed symmetric transfer characteristics with higher currents, indicating the MGr possessed higher carrier injection efficiency for both electrons and holes, which was the reason why the MGr used as the top contact. We also measured the transfer curves of the FET with other bulk or layered metal contacts, but most of these contacts showed an asymmetric carrier injection efficiency and resulted in a stronger p-branch in the $I_{ds}$–$V_g$ curves, as shown in Supplementary Fig. 8. Figure 3c, d show the transfer curves of the bottom-contacted FET at varied $V_{ds}$. The FET with MGr bottom contacts still showed a high and symmetric on-state $I_{ds}$. Reversely, $I_{on}$ of the WTe$_2$-

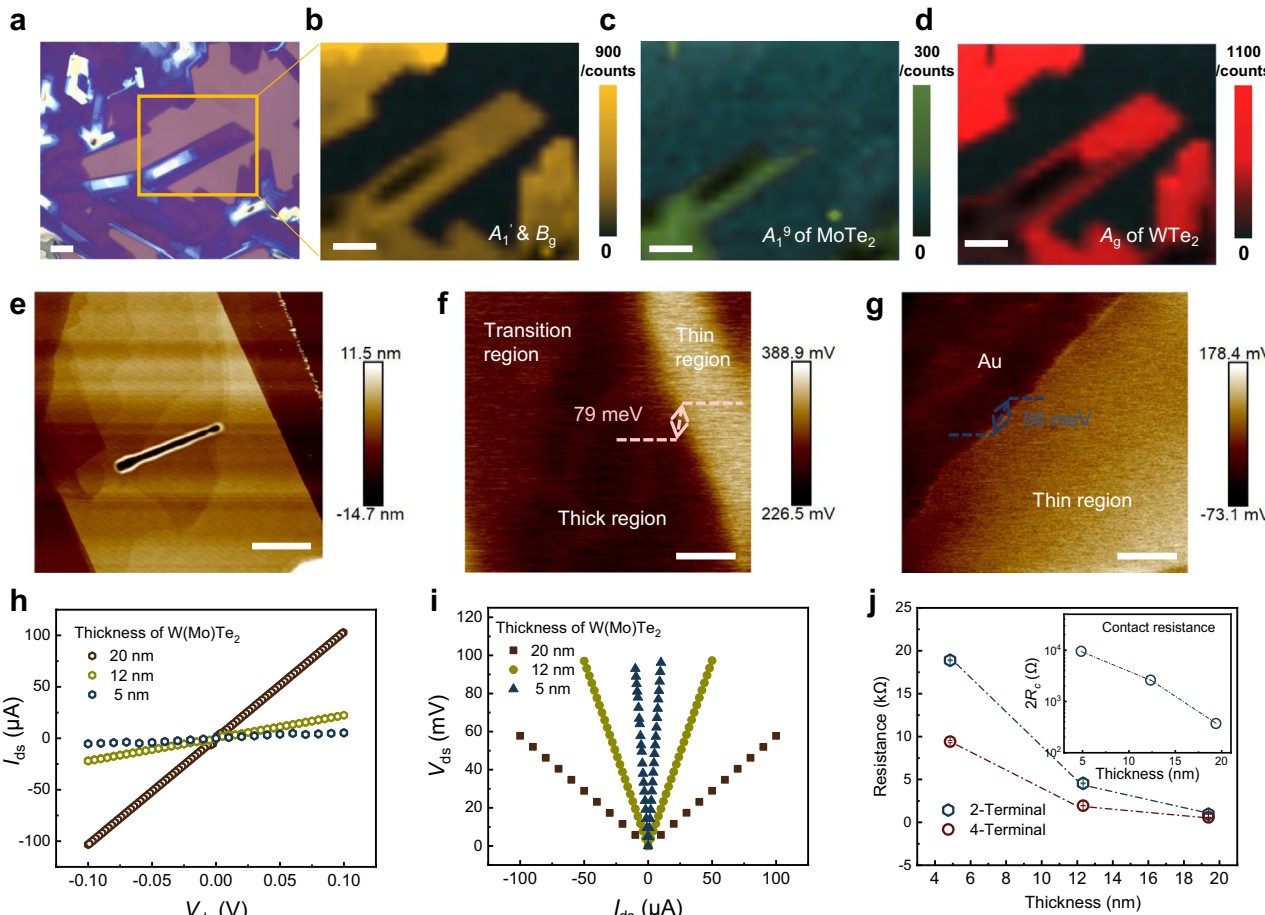

**Fig. 2 | Growth and characterization of WTe₂ bottom electrode. a** Optical image of the epitaxially grown WTe₂. Scale bar: 20 µm. **b–d** Raman intensity mapping images with the characteristic peaks corresponding to $A_1^7$, $A_1^9$ modes of WTe₂, and $B_g$, $A_g$ mode of MoTe₂. Scale bar: 20 µm. **e** Morphology of epitaxially-grown WTe₂ measured by AFM. Scale bar: 4 µm. **f** Potential image of epitaxially-grown WTe₂ measured by Kelvin probe force microscopy. Scale bar: 4 µm. **g** Potential image of Au and CVD-grown WTe₂. Scale bar: 2 µm. **h** I–V curves of W(Mo)Te₂ with various thicknesses measured by the 2-terminal method. **i** I–V curves of W(Mo)Te₂ with various thicknesses measured by the 4-terminal method. **j** 2-terminal resistance ($R_{2T}$) and 4-terminal resistance ($R_{4T}$) of W(Mo)Te₂ with different thicknesses. The standard deviations were used as error bars. The inset shows the contact resistance $2R_{contact} = R_{2T}–R_{4T}$ versus the thicknesses of W(Mo)Te₂.

contacted WSe₂ device was only ~$10^{-9}$ µA/µm at $V_{ds} = 1\,V$, which was six orders of magnitude lower than that of the MGr-contacted FET, the same behavior as shown in the output characteristics (Supplementary Fig. 9b, c). The results suggested that the WTe₂ bottom contacts exhibited a lower carrier injection efficiency due to the self-shielding effect of bottom contact[14], weak interfacial interaction, and vdWs-gap-induced tunneling barrier at the WTe₂/WSe₂ interface.

To explain the contact-geometry-induced suppression of carrier injection, we show the schematic of the current flow pathways in a typical MS surface contact geometry in Fig. 3e–i, where the current flow from metal (A) to channel (D) passes through two regions including the vdWs gap at MS interface (B) and the WSe₂ overlapped with metal (C). We further reduce the surface contact region of WSe₂ FETs into a resistor network under the diffusive approximation, and the contact $R_C$ is expressed in the transmission line model[35]:

$$R_c = \sqrt{\rho_{sc} r_c} \coth(L_c / \sqrt{\rho_{sc} r_c}) \tag{5}$$

where $\rho_{SC}$ is the sheet resistance of the 2D semiconductor beneath the contact, $r_c$ is the specific resistivity of the MS interface, $L_c$ is the contact length, respectively. For the top contact geometry, modulated by the global bottom-gate, $\rho_{SC}$ was decreased as the amplitude of $V_g$ increased, which reduced the $R_C$ and improved the on-state currents. However, for the bottom contact geometry, $\rho_{SC}$ was hardly tuned by

the bottom gate due to the shielding effect of the electrode, resulting in a large contact resistance and a poor on-state current density. The shielding effect of the bottom electrode was also verified by simulation using the COMSOL Multiphysics package, as shown in Supplementary Fig. 10, in which both the electric field and carrier density of the WSe₂ atop the bottom contact were hardly to be modulated by $V_g$. Besides, the vdWs gap between the channel and the vertical side wall of the bottom contact (Supplementary Fig. 10b–o also led to the large and nonadjustable contact resistance due to the large interface resistance $r_c$[36], which was discussed in the previous report[20]. The same tendency could be derived from the schematic energy-band diagrams of the MS structures. As shown in Fig. 3h, the width of the n-type (p-type) Schottky barrier was narrowed as the $V_g$ increased (decreased) in the WTe₂ top contact geometry. In contrast, it is difficult to be modulated in the WTe₂ bottom contact geometry (Fig. 3f).

Apart from the contact geometry, the contact materials were also important. Supplementary Fig. 11 shows the potential difference at the WSe₂/WTe₂, and WSe₂/MGr interfaces measured by KPFM. Compared to the potential difference between WSe₂ and MGr, there existed a smaller potential difference of 37 meV between WTe₂ and WSe₂, indicating that their WFs were horizontally aligned; therefore, WTe₂ had a small charge transfer doping to the WSe₂ and avoided the $\psi_s$ shift of the contacted WSe₂. Meanwhile, for the MGr bottom contact, the electrical contact was dominated by the edge interface of the MGr

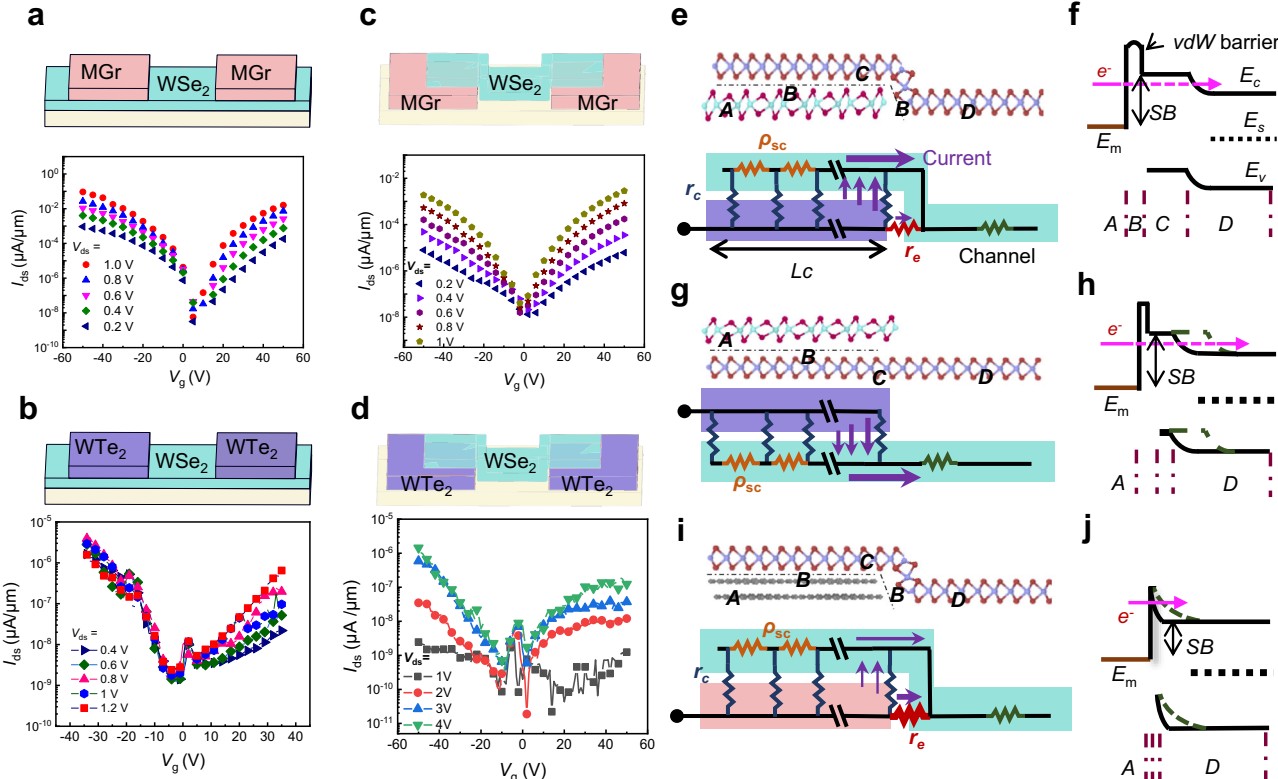

**Fig. 3 | Carrier injection capability of the bottom MGr and WTe₂ contacts.**
**a** Schematic and the transfer curves of MGr top-contacted FET. **b** Schematic and the transfer curves of WTe₂ top-contacted FET. **c** Schematic of and the transfer curves MGr bottom-contacted FET. **d** Schematic and the transfer curves of WTe₂ bottom-contacted FET. **e, f** Schematic of bottom-contacted WTe₂-WSe₂ and the network of contact resistances, and the corresponding schematic of the energy-band diagram. $E_c$, $E_v$, and $E_s$ represent the energies of the conduction band, valence band and Fermi level in 2D semiconductor, respectively. $E_m$ is the work function of the contact metal. SB represents the Schottky barrier. **g, h** Schematic of top-contacted WTe₂-WSe₂ and network of contact resistances, and the corresponding schematic of the energy-band diagram. **i, j** Schematic of MGr-WSe₂ and network of contact resistances, and the corresponding schematic of energy-band diagram. The dashed line in the energy-band diagram indicates the band evolution induced by $V_g$. Brown spheres: selenium atoms. Rose red spheres: tellurium atoms. Green and purple spheres: tungsten atoms. Gray spheres: carbon atoms. Regions A-D represent the metal contact, the interface gap, the contacted semiconductor, and the channel semiconductor region, respectively. $\rho_{sc}$ is the sheet resistance of the semiconductor overlapped with the contact, $r_c$ and $r_e$ are the specific resistivities of the contact gap and edge, respectively, $L_C$ is the contact length.

electrode (Fig. 3j), which was thinner than that of WTe₂ contacts and difficult to suppress the tunneling injecting current. To verify that, the flat-band barrier heights (SHB) of the MGr bottom contact are calculated by 2D thermionic emission mode, as shown in Eq. (6)[9]:

$$I_{ds} = \left[ AA^* T^{1.5} \exp\left(-\frac{q\Phi_B}{k_B T}\right) \right] \left[ \exp\left(\frac{qV_{ds}}{k_B T} - 1\right) \right] \quad (6)$$

where $A$ is the junction area and $A^*$ is the effective Richardson–Boltzmann constant. The obtained $\Phi_B$ as a function of $V_g$ is shown in Supplementary Fig. 9, which indicated that the $I_{on}$ of both p- and n-branch was based on the tunneling mechanism due to the gate-thinned barrier.

The few-layered WTe₂ also showed a weak interfacial interaction with the orbital overlapping to WSe₂, compared to the Au film which possessed a similar WF to WTe₂. We measured the potential difference of WSe₂ on Au and WTe₂ substrate using the WSe₂ on SiO₂ wafer as the reference (Supplementary Fig. 12). The results showed a positive potential difference (~300 meV) between WSe₂ on Au film and the WSe₂ on WTe₂ flake, indicating an unexpectedly strong doping effect of Au film due to the interfacial state, such as metal-induced gap states (MIGS), defect states, and the interface dipoles[37]. Meanwhile, we also compared the intensity and shape-variation of the Raman characteristic peaks of WSe₂ on Au and WTe₂ flake (Supplementary Fig. 13). For

out-of-plane vibrational $A_{1g}$ mode affected by the electrostatic environment change, its full width at half maximum (FWHM) was enlarged as WSe₂ overlapped on Au film compared that of WSe₂ overlapped on WTe₂ (Supplementary Fig. 13c–f), indicating the strong charger transfer doping effect on Au film[38,39]. The detailed comparison is discussed in Supplementary Note 3.

### Reconfigurable unipolar WSe₂ SJFET with asymmetric contact

To suppress the ambipolar behavior while the high on-state performance retained, we fabricated a WSe₂ SJFET with bottom-contacted WTe₂ and top-contacted MGr electrode, as the drain and source contacts, respectively, as shown in Fig. 4a. The optical images of the devices are shown in Supplementary Fig. 9a and the thicknesses of the MGr, WSe₂, and WTe₂ were 13.2 nm, 6.2 nm, and 11 nm, respectively. The transistor characteristics were dominated by both source-drain polarity and control. When $V_{ds}$ was positively biased, as shown in Fig. 4b, the WSe₂ SJFET showed an n-type characteristic, and the on-state current ($I_{on}$) increased to $6 \times 10^{-3}$ μA/μm as $V_{ds}$ increased to 1 V. Meanwhile, the off-state current was suppressed to ~$10^{-10}$ μA/μm at $V_{ds} = 1$ V and $V_g = -60$ V, hence, a maximum on/off ratio higher than $10^6$ was achieved.

When $V_{ds}$ was negatively biased shown in Fig. 4c, the WSe₂ FET displayed a p-type characteristic with $I_{on}$ of $1.1 \times 10^{-2}$ μA/μm at $V_{ds} = -1$V. Similar to the n-type one, $V_t$ increased as the amplitude of $V_{ds}$ decreased, but the leakage current still remained to

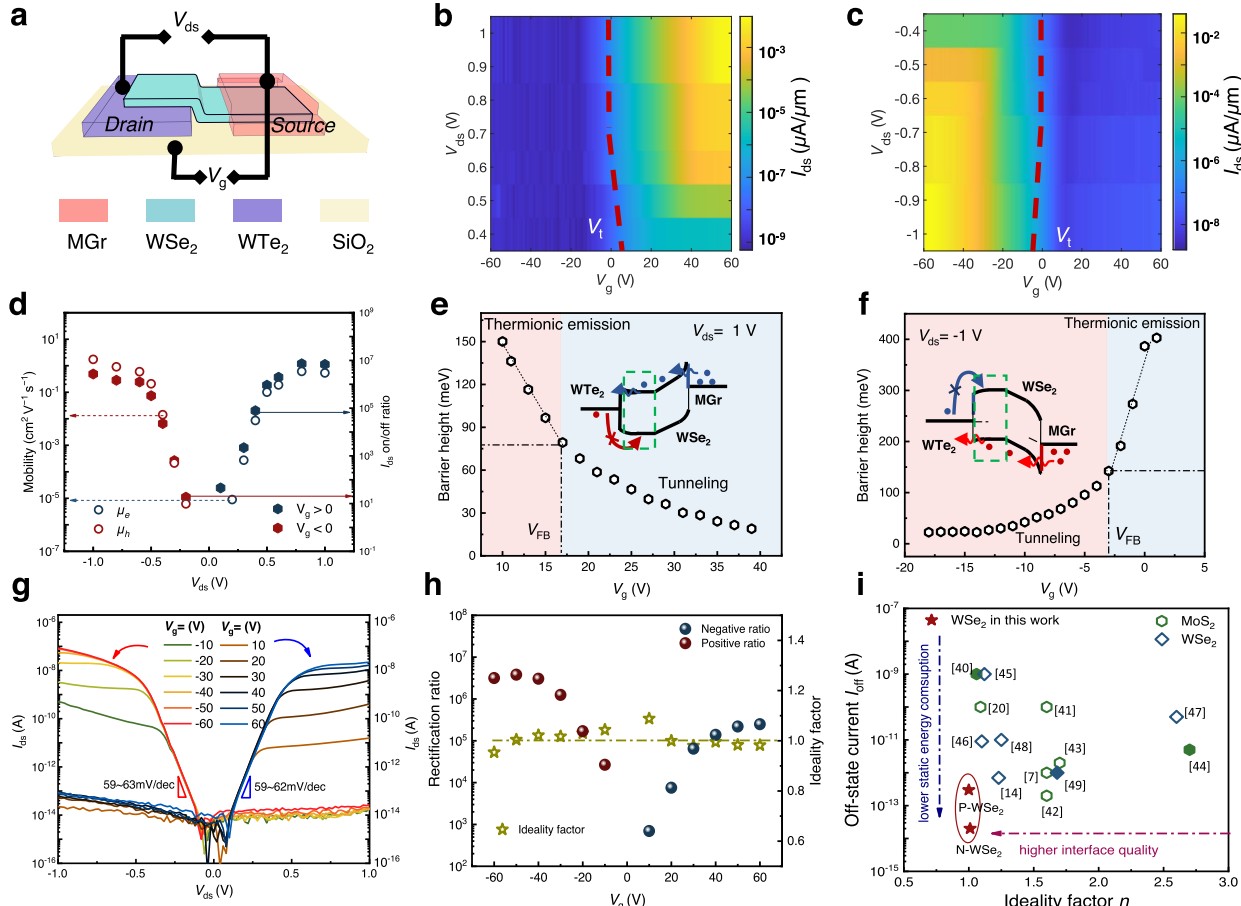

**Fig. 4 | Reconfigurable WSe₂ SJFET with asymmetric MGr/WTe₂ contacts.**
**a** Schematic of asymmetrically contacted WSe₂ SJFET. **b** Pseudo-color transfer plots of the device at $V_{ds} > 0$ showed the n-type polarity. **c** Pseudo-color transfer plots of the device at $V_{ds} < 0$ showed the p-type polarity. The red dash lines indicated the threshold voltage $V_t$. **d** $V_{ds}$-dependent effective field-effect mobility and the $I_{ds}$ on/off ratios. All field-effect mobilities were extracted from the linear regimes. **e, f** Barrier heights of the device at $V_{ds} = 1$ V and $V_{ds} = -1$ V. The Schottky barrier

height is extracted under a flat-band gate voltage ($V_{FB}$) condition, which was responsible for the start of deviations from the linear behavior. **g** Output characteristics of the device at varied gate voltages. **h** Gate-dependent rectifying ratios of the device. The gray-green dashed line represents the mean value of the ideality factors. **i** Comparison of ideality factor $n$ and off-state current of MoS₂[7,21,41–45] and WSe₂[14,46–50] SJFET in previous reports.

be below $1.9 \times 10^{-9}$ μA/μm at $V_{ds} = 1$ V and $V_g = -50$ V. Note that the threshold voltage ($V_t$) shifted with increasing amplitude of $V_{ds}$, because the strong drain electric field penetrated into the channel region and thinned the barrier, resulting in compromised gate-control capability, which was named by the drain-induced barrier lowing (DIBL) effect[40]. The source-drain current $I_{ds}$ on/off ratio at different $V_{ds}$ was summarized in Fig. 4d, showing that the on/off ratio of p-type SJFET was tuned continuously more than $10^6$ and all off-state currents were suppressed at $2.6 \times 10^{-8}$ μA/μm. Meanwhile the on-state $I_{ds}$ of SJFET in the offset geometry were not decreased in comparison with that of WSe₂ FET with the symmetric top contact geometry, as shown in Supplementary Fig. 14.

The off-state leakage power consumption was calculated by $P_{static} = V_{ds} \times I_{ds}$. When $V_{ds} = 1$ V, the $P_{static}$ of the n-type FET was $1.8 \times 10^{-5}$ nW at $V_g = -50$ V with a high on/off ratio, although the p-type FET shows a higher $P_{static}$ of $2.7 \times 10^{-4}$ nW at $V_g = 50$ V. To verify the necessity of the offset contact geometry, we also measured the asymmetric FET in the top contact and bottom geometries, both of which could not simultaneously achieve the ambipolarity to unipolarity conversion (low leakage current) and a high $I_{ds}$ on/off ratio (Supplementary Fig. 15). Meanwhile, we also replaced the bottom electrode with the Au film, as shown in

Supplementary Fig. 16. The bottom-Au-contacted FET showed poor reconfigurability, verifying the WTe₂ indeed played an important role in the polarity control. In addition, the effective two-terminal field-effect mobility ($\mu_{eff}$) for electron and hole on varied $V_{ds}$ were also calculated by Eq. (7):

$$\mu_{eff} = \left(\frac{dI_{ds}}{dV_g}\right)\left(\frac{L}{WC_iV_{ds}}\right) \quad (7)$$

where $V_g$ is the applied back gate voltage and $C_i$ is the capacitance of the SiO₂ dielectric layer (-11.5 nF/cm²). Figure 4d shows that the $\mu_{eff}$ of electron for the WSe₂ FET was almost twice the $\mu_{eff}$ of hole for the device with the MGr contacts. Both $\mu_{eff}$ of electron and hole were strongly influenced by $V_{ds}$ since the calculated effective $\mu_{FET}$ were limited by the contact barrier. To qualify the $V_{ds}$-induced switching of transport polarity, we measured the barrier heights of the asymmetric SJFET at variable temperatures (Supplementary Fig. 17). The $\Phi_{B-n}$ and $\Phi_{B-p}$ of top-MGr contact were obtained from the slope of a linear fit to ln $(I_{ds}/T^{1.5})$ as a function of $1/k_BT$, by employing the 2D thermionic emission Eq. (6). Figure 4e, f show that the $\Phi_{B-n}$ at $V_{ds} = 1$ V was extracted at $V_{FB} = 17$ V to be 79 meV and the $\Phi_{B-p}$ at $V_{ds} = -1$V was extracted at $V_{FB} = -3$V to be 142 meV, although the p-branch $I_{on}$ was slightly higher than the n-branch $I_{on}$. The calculated results

indicated that both p- and n-$I_{on}$ were mainly attributed to the tunneling currents, hence the barrier width instead of height determined the on-state current density.

Figure 4g shows the reconfigurable rectifying behavior of the SJFET with different $V_g$ in which the rectifying direction was switched by $V_g$. The maximum rectifying ratio reached $3\times10^6$ in the positive rectifying direction and $2.5 \times 10^5$ in the negative rectifying direction, whereas the rectifying ratio of the all-MGr-contacted WSe$_2$ Schottky diode was only about 10 at $V_{ds} = \pm1\,V$. To further evaluate the rectifying performance of the gate-tunable WSe$_2$ Schottky junction diode, an ideality factor ($n$) was estimated at a small forward bias (here is 0.02–0.35 V) by fitting to Schottky diode Eq. (8).

$$I_{ds} = I_s \left[ \exp\left(\frac{V_{ds}}{nV_T} - 1\right) \right] \qquad (8)$$

where $I_{ds}$, $I_s$, $V_{ds}$, and $V_T$ denote the drain current, reverse leakage current, drain voltage, and thermal voltage, respectively. As the gate voltage swept from positive to negative in Fig. 4h, the ideality factor $n$ derived from the parameters of the fitting equation was nearly fixed on 1 with negligible variation, indicating a near-ideal diode attribute in the reconfiguration process. Figure 4i summarizes the reported ideality factors and off-state current of the 2D SJFET, indicating the high quality of the asymmetric contacted SJFET and the application potential towards lower static power dissipation.

The reconfigurable rectifying operation was based on unpinned energy level at the MGr/WSe$_2$ interface and the strong carrier-injection suppression capability of WTe$_2$. As shown in Supplementary Fig. 18, when $V_g > 0$ at $V_{ds} > 0$, the gate-electric field induced strong electron accumulation and reduced the $\psi_s$ of the MGr-contacted WSe$_2$. Hence the SB width was thinned to promote the electron injection from MGr through the DT (Supplementary Fig. 18b). In contrast, when $V_g < 0$, the width of barrier at WTe$_2$/WSe$_2$ interface remained constant due to the shielding effect, which reduced the off-state hole current leakage (Supplementary Fig. 18c). Reversed carrier injection process happened at $V_{ds} < 0$ (Supplementary Fig. 18d–f), only holes were allowed to be injected from the MGr side when $V_g < 0$. The WTe$_2$ contacting strategy can also be applied to fabricate the reconfigurable MoS$_2$ SJFET (Supplementary Fig. 19). The SJFET with Au/WTe$_2$ contacts showed a gate-tunable rectifying characteristic with rectification ratios ranging from 1 to $10^5$. Compared to similar transport curves of Au/MGr contacted SJFET at $V_{ds} = \pm1\,V$ (Supplementary Fig. 19d and e), the transport behavior of Au/WTe$_2$ contacted device was determined by the sign of $V_{ds}$, because the electron injection from WTe$_2$ was inhibited, as shown in Supplementary Fig. 19c.

### Gate-tunable photo-response of the SJFET

Because the SJFET is regarded as equivalent to two back-to-back Schottky junctions at the asymmetric contact interfaces, the photo-response was tuned by both the $V_{ds}$ and the $V_g$. We used two devices to investigate the photo-response and the optical image are shown in Supplementary Fig. 10a, b. The data in Fig. 5a–c were derived from sample 2#. We first investigated the photocurrent $I_p$ and photo-responsivity at positive and negative $I_{ds}$ when $V_g = 0$ (Supplementary Fig. 20), which showed a nearly linear increase with the laser power intensity. More details of power-dependent photo-response are shown in Supplementary Note 4. More importantly, the SJFET also showed a potential as a self-powered photodetector due to its tunable photo-voltaic performance. Supplementary Fig. 20c shows the power-dependent temporal short-circuit current $I_{sc}$ at $V_g = 0$. The $I_{sc}$ was slightly lower than the photocurrents at the same power density, but the photovoltaic response had a smaller dark current and a lower power consumption since $V_{ds}$ was not required.

The photovoltaic responses of the WSe$_2$ FET were further tuned by $V_g$. Figure 5a, b shows the output curves on varying positive and

negative $V_g$ under the same laser power series ($P_{in}$ = 3.5 mW/cm$^2$, and a wavelength of 635 nm). With $V_g$ positively increasing, $I_{sc}$ and open-circuit $V_{oc}$ gradually entered the saturated region. The negative $V_g$ modulated behavior was similar. The gate-tunable $V_{oc}$ and $I_{sc}$ were summarized in Fig. 5c, showing that the $V_{oc}$ was tuned from 0.29 V to −0.47 V and the self-powered responsivity $R_{sc}$ was tuned from −12.7 mA/W to 61.7 mA/W. Figure 5d, e shows that regardless of the positive and negative gate bias, $I_{sc}$ and output electrical power density showed an exponential increase with light power intensity, and $V_{oc}$ also monotonically increased (Fig. 5f). Hence the power conversion efficiency $\eta_{PV}$ calculated by $\eta_{PV} = P_{out}/P_{photo}$ was almost fixed at 0.37% at $V_g = 60\,V$ and 0.15% at $V_g = -60\,V$, respectively, although the incident power increased by two orders of magnitude. Although the output electrical power density and $V_{oc}$ were effectively modulated through the gate control under varying laser power density, $I_{sc}$ showed a weak gate-tunable capability, in which the corresponding self-powered responsivity reached 30 mA/W at $V_g > 10\,V$ and 1 mA/W at $V_g < -10\,V$, respectively. The filling factor (FF) qualifies how closely a photovoltaic device acts as an ideal source. Figure 5g, h show the gate-modulated FF of the SJFET at varying incident power density due to the change of $V_{oc}$. As the amplitude of $V_g$ increased, the SJFET yielded an increased FF, reaching 0.60 at $V_g = -60\,V$ and 0.68 at $V_g = 60\,V$. Supplementary Table 1 shows a photovoltaic performance comparison among the asymmetric contacted WSe$_2$ SJFET and the previously reported photovoltaic devices, which implies the high photovoltaic performance of WSe$_2$ SJFET at both positive and negative $V_g$. The reversible photovoltaic performance rendered the asymmetric SJFET to work as the self-powered logic inverter at an ambient light level, as shown in Fig. 5i and Supplementary Fig. 20d, with $V_g$ as the input signal and $V_{oc}$ as the output signal. Even at low illuminance level ($P_{in} = 0.1$ mW/cm$^2$), the logic inverter still showed the obvious $V_{oc}$ switch from −0.2 V to 0.28 V, which further decreased the static power dissipation in integrated circuits due to null $V_{ds}$ applied on the SJFET.

## Conclusions

In conclusion, we proposed a contact-engineered SJFET with the reconfigurable polarity and low leakage current, achieved by employing the asymmetrically vdWs semimetal contacts in which the carriers were only injected from the MGr contact and the injection was suppressed at the epitaxially-grown WTe$_2$ bottom contact. The asymmetrically contacted WSe$_2$ SJFET in the offset geometry showed the conversion between ambipolarity and unipolarity and the alternative carrier polarity was determined by the drain bias. Meanwhile, the leakage currents were effectively suppressed to $2 \times 10^{-9}$ μA/μm and the device showed a controllable $I_{ds}$ on/off ratio with a maximum of $10^6$. The off-state leakage power consumption was reduced to $10^{-5}$ nW (n-type) and $10^{-4}$ nW (p-type) at $V_{ds} = \pm1\,V$. Also, the WSe$_2$ SJFET also exhibited a reversible rectifying behavior with a maximum rectifying ratio of $3 \times 10^6$ and an ideality factor of 1. Advantageously from the electrically gate-tuned contact barrier, the drain-engineered SJFET exhibited a runtime reversible photovoltaic performance in which the sign of the photo-responsivity was substantially tuned and the $V_{oc}$ was switched markedly between the −0.47 V and 0.29 V. Furthermore, based on the photovoltage-reversible properties of the photodiode, a logic optoelectronic device was designed to realize the switch between positive situation to negative situation by manipulating the gate voltage. This contact engineering strategy is generally applicable to other 2D materials such as the electrically gate-tunable n-type MoS$_2$ Schottky diode. The modulation of carrier injection in 2D materials also provides an alternative route to reduce the logic-circuit complexity and promises innovation for the future applications of computational sensors and optical communications.

Note: during revision of this manuscript, we became aware of a related work[20].

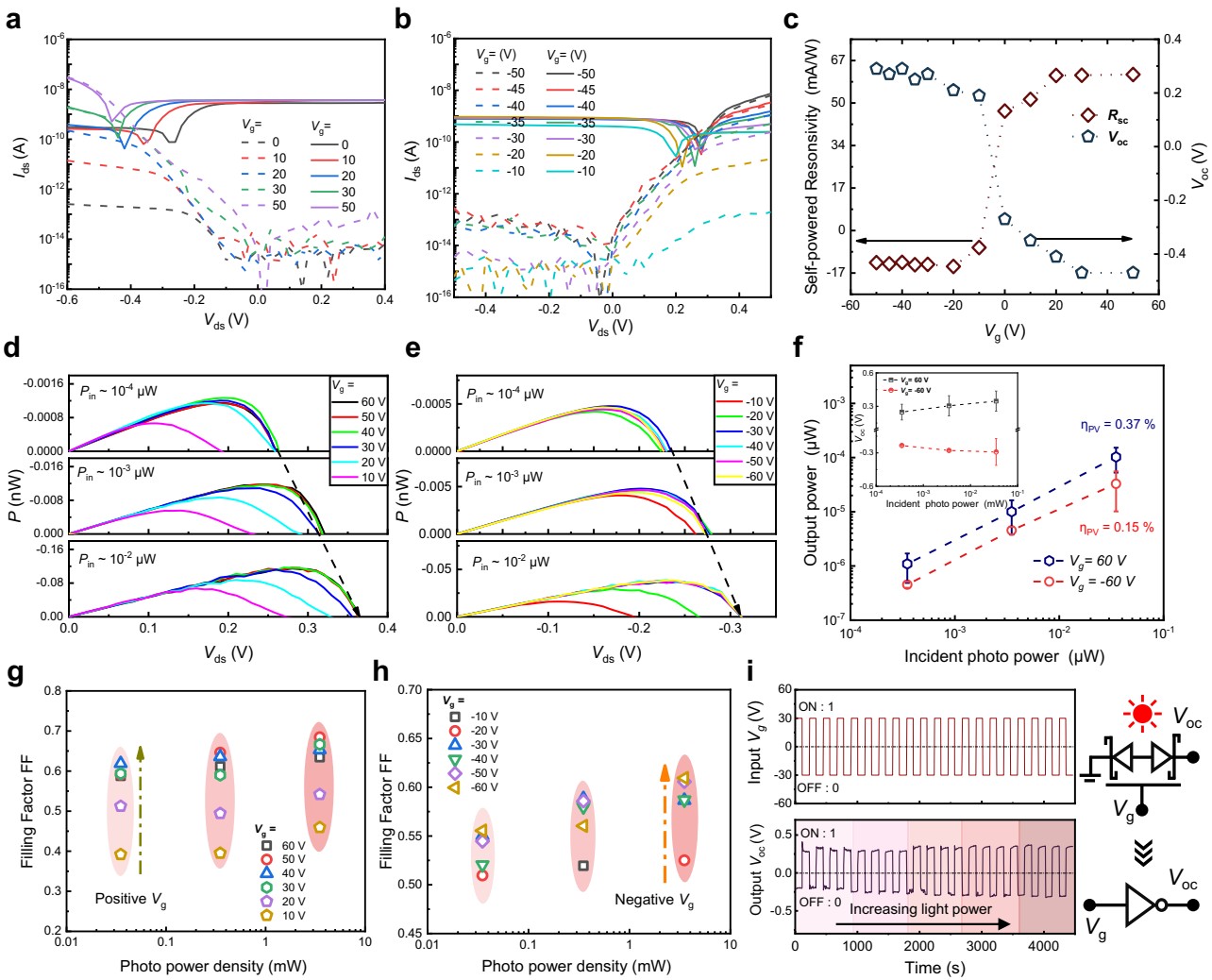

**Fig. 5 | The photovoltaic performance of asymmetric contacted SJFET. a, b** Gate-dependent output curves of the device under illumination (635 nm). Dashed lines were measured in the dark. **c** Gate modulation of the responsivity ($R_{sc}$) at $V_{ds} = 0$ V and open-circuit voltage $V_{oc}$, respectively. **d, e** Output electrical powers at $V_g > 0$ and $V_g < 0$ as a function of drain-source voltage, respectively. $P_{in}$ is the incident-light intensity. The dashed line indicates the increasing trend of $V_{oc}$ with $P_{in}$. **f** Output electrical power at $V_g > 0$ and $V_g < 0$ as a function of incident-light density.

The inset shows $V_{oc}$ vs incident power density. the incident power density. $\eta_{PV}$ is the power conversion efficiency. The standard deviations were used as error bars. **g, h** Fill factor (FF) at $V_g > 0$ and $V_g < 0$ as a function of the incident power density. The FF increased with the increasing amplitude of $V_g$. **i** Logic inverter based on the gate switchable photovoltaic performance. The white light power intensity was ranged from 0.1 mW/cm² to 30 mW/cm².

## Methods

**One-step epitaxial growth of WTe₂.** The molten-salt-assisted thermal chemical vapor deposition (CVD) method was used to synthesize WTe₂. A mixture of 20 mg hydrate $(NH_4)_6Mo_7O_{24}\cdot4H_2O$, $(NH_4)_{10}W_{12}O_{41}\cdot xH_2O$ (Sigma-Aldrich) and sodium cholate (Sigma-Aldrich) in a mass ratio of 5:5:1 and the SiO₂ substrate was placed in the middle of the heating zone, with a Te lump placed 1 cm away from the substrate. Throughout the growth process, a carrier gas mixture of H₂/Ar at a flow rate of 10/100 sccm was utilized. The temperature of the heating zone gradually increased to 760–860 °C and held for 3–5 min. By using these two mixed hydrates as the precursor, the MoTe₂/WTe₂ semimetal heterostructures were epitaxially synthesized in a one-step method in which the thicker MoTe₂ flakes were synthesized firstly, then the WTe₂ epitaxially were grown along the edges of MoTe₂. As the reaction time increased, the interspaces of MoTe₂ frameworks were covered with the polycrystalline WTe₂ to form a continuous MoTe₂/WTe₂ film, as shown in Supplementary Fig. 2e. More information about the sample growth is detailed in Supplementary Note 1.

**Material characterization.** AFM (Bruker, Dimension Icon) in the tapping mode TUNA mode were employed to measure the thickness of device, while the contact potentials of the different areas were measured via the Kelvin probe force microscopy. Micro-Raman investigation was performed using HORIBA LabRAM HR Evolution system with 532 nm laser excitation (the laser spot was ~1 μm). The morphology and chemical composition distribution of WTe₂/MoTe₂ were analyzed by SEM, and XPS (Thermo Fisher Scientific, K-Alpha+). The crystal structure of 2D flakes was characterized by the TEM (FEI Tecnai F200 systems) operated at 80 kV. The TEM sample was prepared using PMMA-supported wet-transfer method.

**Electrical characterization.** The SJFET devices were tested in a Cascade probe station under high vacuum conditions. The electrical measurement was performed through the Keithley 4200 semiconductor characterization system. Electrical conductivity measurements were taken from 340 K to 80 K with a cooling rate of 2 K/min. The dwell time at each test temperature was 10 min. The 635 nm lasers were used for light illumination and controlled by the Thorlabs ITC

4001. The power density was 30 mW/cm². During the measurements, the devices were positioned at the center of the light spot.

## Data availability

Relevant data supporting the key findings of this study are available within the article and the Supplementary Information file. All raw data generated during the current study are available from the corresponding authors upon request.

## Code availability

The code that supports the findings of this study is available from the corresponding author upon request.

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

## Acknowledgements

The work is in part supported by the Research Grants Council of Hong Kong, particularly, via Grant AoE/P-701/20, 14206721, National Natural Science Foundation of China, Grant Nos. 62005051, 62104165 and 62274114. The Natural Science Foundation of Jiangsu Province, Grant No. BK20210713. Gusu Youth Leading Talent, Grant No. ZXL2021452, RGC Postdoctoral Fellowship, CUHK Group Research Scheme, CUHK Postgraduate Studentship, CUHK Postdoctoral Fellowship, CUHK Fund for Joint Research Labs.

## Author contributions

Y.Z. and J.-B.X. conceived the idea and designed the experiments. J.-B.X. supervised the whole project. Y.Z. carried out device fabrication, measurements, and analysis. Z.C. conducted the numerical simulation. L.Tao and Y.P. contributed to the discussions. Y.Z., L.Tong, Z.C., and J.-B.X. contributed to the discussions and co-wrote the manuscript with input from all the co-authors.

## Competing interests

The authors declare no competing interests.
