## [Peer Review File · Nature Communications]

Contact-engineered reconfigurable two-dimensional Schottky junction field-effect transistor with low leakage currentsREVIEWER COMMENTS

Reviewer #1 (Remarks to the Author):

The authors reported a reconfigurable Schottky junction transistor that possessed a very low I_{ds} leakage energy consumption (around 10^{-10} $\mu\text{A}/\mu\text{m}$) and meanwhile kept a high I_{ds} on/off ratio ($> 10^6$), by using the exfoliated graphite and epitaxially grown WTe₂ as the offsetting contacts. The authors discussed the mechanism of ambipolarity to unipolarity conversion by designing contact structures and materials. The polarity of the SJFET can be switched by the V_{ds} and showed an ideal rectifying behavior with the global gate control. Finally, the author showed the photovoltaic application of SJFET, such as the photo-powered logic inverter. The experimental results in this paper are benchmarking. I would like to recommend it for consideration for publication after a major revision following the technical points that should be further clarified, as the high standards of Nature communications are required.

1. The thickness of the graphene, WTe₂, and WSe₂ should be measured and clarified clearly. Usually, the monolayer graphene, as the Dirac metal, was used as the reconfigurable contact due to its gate-modulating Fermi level. However, if the graphene is multilayered, the conductance is hard to be modulated. Also, this asymmetric contacted SJFET seems to be regarded as a semi-gate transistor (e.g. Nat. Electron. 5, 752-760 (2022)) in which the WTe₂ was regarded as the shielding layer. Hence, the thickness of WTe₂ is also important, and its shielding effect should be clarified.
2. The author should explain why line-scanning Raman spectra in figure 2e showed a gradual transition from WTe₂ to MoTe₂, considering that the sample has a sharp WTe₂/MoTe₂ interface.
3. It is necessary to add the reference about the polarization angle of Raman modes (A_{1g}) of WTe₂ and (A_g) of MoTe₂ to indicate the consistency of growth direction.
4. Figures 4b and 4c should be explained more sufficiently. For example, the V_{ds} strongly affect the shifting of the threshold voltage. The author should give more explanation.
5. The references in Figure 4 should be labeled at corresponding positions in Figure 4i.

Reviewer #2 (Remarks to the Author):

In the manuscript, the authors reported the reconfigurable asymmetric Schottky junction FETs using an ambipolar WSe₂ channel and the vdW metallic WTe₂ contact. In addition, they investigated the gate-controlled polarity conversion properties and their utilization as a visible-light photodetector. To achieve the ambipolar-to-unipolar polarity conversion while suppressing the leakage current, the authors employed the electrostatically tunable graphene and electric-field-shielding WTe₂ as top and bottom contacts to the WSe₂ channel with the laterally contacted FET geometry, respectively. The authors have also performed intensive electrical and optoelectrical characterization for the suggested device structure. However, despite the demonstration of high performance for rectification ratio and off-state current than previous reports, I could not find any other remarkable result or significant advances or structural novelty in this work, considering that many TMD-heterojunction-based tunable rectifiers or photodiodes with similar device geometries have been reported.

1. The authors demonstrated reconfigurable WSe₂ SJFET with effectively suppressed leakage current. However, I am not sure it presents methodological advances considering the previous results. The previous method also shows a "shielding effect" via bottom vdW contact, demonstrating a reconfigurable rectifier with low off-state leakage current $10^{-14} \sim 10^{-12}$ A at $V_{ds} = 1$ V and gate-tunable photovoltaic property with V_{oc} of ~ 0.8 V. (Nano Energy 49 103-108 '18, J. Mater. Chem. C, 6(43), 11673-11678 '18).
2. In the suggested device structure, what are the advantages of using the WTe contact instead of metal electrodes? Schottky contacted metals can shield the gate electric field more effectively, and

the Fermi-level depinning is also achieved by vdW transferred metal contacts, according to many recent reports. Additionally, it makes device fabrication much easier.

3. In Figure 3, the charge injection ability of the graphene bottom electrode was compared with the bottom electrode WTe₂. But is there any reason why graphene was used as the top electrode in the final device structure? In this case, carrier injection through top graphene contact might not be efficient enough because channel WSe₂ multilayers may shield the portion of the electric field.

4. In Fig. 4, the authors demonstrated reconfigurable unipolar behavior in WSe₂ SJFETs with asymmetric MGr/WTe₂ contacts. In this device structure, the authors claimed that WTe₂ shields the gate electric field effectively so that carriers cannot overcome the SBH at the junction between WSe₂ and WTe₂. However, considering the bottom contact geometry, the WSe₂ channel material at the junction on the edge of WTe₂ is affected by the gate, and the transport behavior is likely to differ.

Some other technical comments:

- In Fig. 2 i-k, it needs to specify the contact geometry. For a fair comparison, 4-terminal conductance should be measured and compared.
- Field-effect mobility extraction may not be appropriate using equation (3) because the Schottky barrier will also vary with gating.
- Output characteristics of the device at variable voltage are investigated in Figure 4 g and Figure 5 a, b. However, the gate-dependent rectification properties of the WSe₂ diode are inconsistent with each other for the dotted line representing dark current in Figure 4 g and Figure 5 a, b. As the author indicated the WTe₂ as a drain contact, it would be better to correct the results in Figure 5 a, b.

Reviewer #3 (Remarks to the Author):

In this manuscript, the authors designed an offset Schottky junction field-effect transistor with asymmetric contact of semimetal graphene and WTe₂, where they are top and bottom contact, respectively. They claimed that the design can suppress the off-state current and control the polarity of the transistor. After carefully reading the manuscript, I think the manuscript can not be accepted for publication at such situation.

1. Most of the pictures are blurred and difficult to see clearly. Maybe high resolution pictures are better.
2. Lots of errors make the readability of the manuscript not good. What does the sentence (line 20-26) mean? Figure 1a: symmetric or asymmetric (line 134)? Figure S8: WSe₂ or MoS₂? Line 266: "However, the off-state current was suppressed to 10-10 $\mu\text{A}/\text{cm}^{-1}$ even at $V_{\text{ds}} = 1\text{V}$ and $V_{\text{g}} = -60\text{V}$.", and more.
3. What is the unit of conductivity, mS or S/m? The authors claimed that WTe₂ possesses high conductance and appropriate work function to enhance the shielding effect in comparison with graphene. Can the authors give the conductivity of their graphene to support their conclusion?
4. Following comment 3, maybe the polarity control is mostly from the asymmetric work function of graphene and WTe₂ like asymmetric metal electrodes? The authors should compare their offset device with devices of asymmetric contact, both bottom or top.
5. Maybe moving the part of WTe₂ growth to the supplementary materials is better.
6. The authors should provide pictures of their devices, which may help readers well understand their work.
7. The authors should give more detailed explanation on the working principle of the offset device, like combining schematic pictures, which should help readers to well understand the device.
8. The authors claimed that the structure can suppress the off-state current. From the manuscript, it can be seen that the on/off ratio is not high. Does the structure also suppress the on-state current? This is not good for device applications. How does it compare with metal electrodes like Adv. Funct. Mater. 31, 2007559 (2021)?
9. Figure 3 is for symmetric contact devices. Theoretically, the transport should be independent on the direction of V_{ds} . In Figure 3c and 3d, the results exhibit they are dependent. Is this from the difference of the source and drain? I mean maybe the contact area, thickness and others are different and affect the device performance.

**Response to Reviewers' Comments**

**Response to Reviewer 1#**

**General comments:**

*"The authors reported a reconfigurable Schottky junction transistor that possessed a very low*
*I_{ds} leakage energy consumption (around 10^{-10} $\mu A/\mu m$) and meanwhile kept a high I_{ds} on/off ratio*
*($> 10^6$), by using the exfoliated graphite and epitaxially grown WTe_2 as the offsetting contacts.*
*The authors discussed the mechanism of ambipolarity to unipolarity conversion by designing*
*contact structures and materials. The polarity of the SJFET can be switched by the V_{ds} and*
*showed an ideal rectifying behavior with the global gate control. Finally, the author showed*
*the photovoltaic application of SJFET, such as the photo-powered logic inverter. The*
*experimental results in this paper are benchmarking. I would like to recommend it for*
*consideration for publication after a major revision following the technical points that should*
*be further clarified, as the high stands of Nature Communications are required."*

**Response:**

We thank the reviewer for the valuable comments and suggestions. In the revised
manuscript, we added experiments, simulations, and discussions to verify the working
principles of the SFFET with offset contact geometry and some details of epitaxial growth of
the WTe_2 according to the reviewer's comment. **We have responded point by point to the**
**reviewers' concerns, added revisions, and highlighted them in the manuscript,** and hope that the
revised manuscript will address all your concerns.

**The main responses are summarized as follows:**

- ➤ We specified the thickness of the multi-layer graphene, WSe_2 , and WTe_2 . We also added
more explanations and figures on the working principle of SJFET with offset contact
geometry.
- ➤ We used electrical characterization and COMSOL simulation to show the shielding effect
of the WTe_2 bottom contact.
- ➤ We used the phase contrast image measured by atomic force microscope (AFM) to the sharp
interface in $W(Mo)Te_2$ and explained the reason for the gradual transition in line-scanning
Raman spectral.
- ➤ We added more explanation on the V_{ds} -induced threshold voltage shifting, attributed to the
drain-induced-barrier-lowering effect.
- ➤ We added the corresponding references to confirm the epitaxial growth of WTe_2 . We also
labeled the reference number in the corresponding location in **Figure 4i**.

**Comment 1:**

*"The thickness of the graphene, WTe_2 , and WSe_2 should be measured and clarified clearly.*
*Usually, the monolayer graphene, as the Dirac metal, was used as the reconfigurable contact*

*due to its gate-modulating Fermi level. However, if the graphene is multi-layered, the*
*conductance is hard to be modulated. Also, this asymmetric contacted SJFET seems to be*
*regarded as a semi-gate transistor (e.g. Nat. Electron. 5, 752-760 (2022)) in which the WTe₂*
*was regarded as the shielding layer. Hence, the thickness of WTe₂ is also important, and its*
*shielding effect should be clarified.”*

**Response:**

We thank the reviewer for the valuable comments and suggestions. We have specified the
thickness of each layer in our device in the revised manuscript. The thickness of multilayer
graphene (MGr), WTe₂, and WSe₂ is 13.2 nm, 6.2 nm, and 11 nm respectively as shown in
**Figures S11c-d (Figures R1a-b)**. As the reviewer’s comments, the Fermi level of MGr cannot
be modulated. Hence, in the SBFET, the reconfigurable property originated from the
electrostatic doping of the WSe₂ channel. Since the work function of MGr was almost aligned
horizontally to the intrinsic Fermi level of the WSe₂, both hole and electron were injected into
the WSe₂ channel through the coordinated regulation of the gate voltage V_g and drain voltage
V_{ds} .

To verify the shielding effect of WTe₂, we measured the transfer curves of the WTe₂ and MGr
contacted WSe₂ in top contact and bottom contact geometry, respectively, as shown in **Figures**
**3a-d (Figures R1c-f)**. Compared to the MGr bottom-contacted device, the WTe₂ bottom-
contacted SJFET showed much lower on-state I_{ds} at $V_{ds}=1V$ due to the shielding effect of the
bottom electrode.¹ We also used the COMSOL Multiphysics package to demonstrate the
shielding effect of the WTe₂ bottom electrode and the results are shown in **Figure S10 (Figures**
**R1 g-j)**. In contrast to the WSe₂ under the top contact, the electric field of the WSe₂ upper the
bottom contact was hard to be modulated by the V_g .

**Figure R1. a)** AFM height image of MGr/WSe₂/WTe₂ SJFET. Scale bar 2 μm . **b)** Height profile of the
 MGr, WSe₂, and WTe₂ measured along the dash lines in **Figure R1a**. **c) and d)** Transfer curves of
 MGr/WSe₂/WTe₂ SJFET in top and bottom contact geometry. **e) and f)** Transfer curves of
 WTe₂/WSe₂/WTe₂ SJFET in top and bottom contact geometry. **g)-h)** Distribution of electric field
 through the WSe₂ with top WTe₂ contacts, at $V_g = 0$ and 100 V, respectively. **i)-j)** Distribution of
 electric field through the WSe₂ with bottom WTe₂ contacts, at $V_g = 0$ and 100 V, respectively. Scale
 8 bar: 5 nm.

**The corresponding discussions in the revised manuscript:**

“The photograph of the device is shown in **Figure S9a** and the thickness of MGr, WSe₂, and WTe₂ is
 13.2 nm, 6.2 nm, and 11 nm, respectively.” (Page.12, line 3)

“**Figures 3a and 3b** show the schematics and transfer curves of the top-contacted device at varying
 V_{ds} ... The results suggested that the WTe₂ bottom contacts exhibited a low carrier injection efficiency due
 to the self-shielding effect of bottom contact,¹⁴ weak interfacial interaction, and vdWs-gap-induced
 tunneling barrier at the WTe₂/WSe₂ interface.” (Page.9, line 4)

“The shielding effect of the bottom electrode was also verified by simulation using the COMSOL
 Multiphysics package as shown in **Figure S10**, in which the electric field and carrier density of the
 WSe₂ upper the bottom contact was hard to be modulated by the V_g ... also led to the large and
 nonadjustable contact resistance due to the large interface resistance r_c , which was discussed in the
 previous report²⁰.” (Page.9, line 29)

“Figure S10. Distribution of electric field and carrier density through the SJFET with different geometries

simulated using the COMSOL Multiphysics package.” (Supporting information, Page.15, line 2)

**Reference:**

1. Du, J. *et al.* Gate-Controlled Polarity-Reversible Photodiodes with Ambipolar 2D Semiconductors.
*Adv. Funct. Mater.* **31**, 2007559 (2021).

**Comment 2:**

“The author should explain why line-scanning Raman spectra in figure 2e showed a gradual
transition from WTe_2 to $MoTe_2$, considering that the sample has a sharp $WTe_2/MoTe_2$ interface.”

**Response:**

We thank the reviewer for the valuable suggestions. The resolution of line-scanning Raman
spectra and Raman intensity mapping was limited by the laser spot size (The diameter of the
spot size is around 1.5 μm). When the laser was irradiated on the interface of $MoTe_2$ and WTe_2 ,
the Raman signals from both parts were collected, which induced the gradual transition in the
line-scanning Raman spectra ¹.

Another possible reason for the gradual transition is that the location of the $MoTe_2$ to WTe_2
interface is layer dependent. We used phase-contrast imaging in atomic force microscopy (AFM)
measurement to further characterize the interface of $MoTe_2/WTe_2$ at each layer. We did not
observe a clear interface in both the optical and the height images (**Figures R2a-b**). However,
due to the variation in tip-sample dissipation over the $MoTe_2$ and WTe_2 region, the $MoTe_2$ and
WTe_2 showed different phase shifts in the phase contrast image as shown in **Figure S4c** (**Figure**
**R2c**). The result showed that the location of $MoTe_2/WTe_2$ was closer to $MoTe_2$ nucleation with
the increased thickness although the interface at each layer was sharp.

**Figure R2. a)** Photograph of the epitaxial-grown $W(Mo)Te_2$. Scale bar 10 μm . **b)** Height profile of the
epitaxial-grown $W(Mo)Te_2$ measured by AFM. **c)** Phase contrast image of the epitaxial-grown $W(Mo)Te_2$.

**The corresponding discussions in the revised manuscript:**

“When the laser was irradiated on the interface of $MoTe_2$ and WTe_2 , the Raman signals from both
parts were collected, ...which indicated this sharp interface at the position of the thickness step.”

(Supporting information, Page.12, line 3)

**Reference:**

1. Chen, K. *et al.* Lateral Built-In Potential of Monolayer MoS₂-WS₂ In-Plane Heterostructures by a
Shortcut Growth Strategy. *Adv. Mater.* **27**, 6431–6437 (2015).

**Comment 3:**

“It is necessary to add the reference about the polarization angle of Raman modes (A_{19}) of WTe₂
and (A_g) of MoTe₂ to indicate the consistency of growth direction.”

**Response:**

We thank the reviewer for the valuable suggestions. We have added references ^{1,2} (*Adv.*
*Funct. Mater.* 2017) to indicate that the polarization direction of WTe₂ (A_{19} mode at 210 cm⁻¹)
was parallel to the *a*-axis of the WTe₂ crystal. Also, we have added references ^{3,4} (*J. Phys. Chem.*
*C* 2020) to indicate that the polarization direction of MoTe₂ (A_g mode at 160 cm⁻¹) is
perpendicular to the *a*-axis of the MoTe₂ crystal. These results are consistent with our results as
shown in **Figures S4d-f (Figure R3)**.

**Figure R3. a)-b).** Polarized Raman spectra of WTe₂ and MoTe₂ measured with incident laser rotation
configuration. The dashed line indicates the direction of the *a*-axis of the WTe₂ and MoTe₂ crystals.
**c)** Top views of lattice structures of monolayer T_d phase WTe₂ and MoTe₂. Brown spheres: tellurium
atoms. Blue spheres: W atoms. Green spheres: Mo atoms.

**The corresponding discussions in the revised manuscript:**

“Raman modes (A_{19}) of WTe₂ and (A_g) of MoTe₂ are linearly polarized and show a consistent two-fold
symmetric intensity curve ^{3,4}” (*Supplementary Information Page.12, line 3*)

**References:**

1. Ye, X.-G. *et al.* Control over Berry Curvature Dipole with Electric Field in WTe₂. *Phys. Rev. Lett.* **130**,
16301 (2023).

2. Song, Q. *et al.* The polarization-dependent anisotropic Raman response of few-layer and bulk WTe₂
under different excitation wavelengths. *RSC Adv.* **6**, 103830–103837 (2016).

3. Beams, R. *et al.* Characterization of Few-Layer 1T' MoTe₂ by Polarization-Resolved Second
Harmonic Generation and Raman Scattering. *ACS Nano* **10**, 9626–9636 (2016).

4. Wang, J. *et al.* Determination of Crystal Axes in Semimetallic T'-MoTe₂ by Polarized Raman
Spectroscopy. *Adv. Funct. Mater.* **27**, 1604799 (2017).

**Comment 4:**

“Figures 4b and 4c should be explained more sufficiently. For example, the V_{ds} strongly affect
the shifting of the threshold voltage. The author should give more explanation.”

**Response:**

We thank the reviewer for the valuable suggestions. We added more explanation of the
shifting of the threshold voltage (V_{th}). This V_{th} shifting fundamentally originated from the
competitive control of gate voltage V_g and source-drain bias V_{ds} ¹. When V_{ds} increased, electric
fields created by the drain were stronger to penetrate into the channel region and thinned the
barrier, resulting in compromising the gate control capability over the channel as shown in
**Figure R4.**

**Figure R4.** Simulation of electron potential energy in the channel region via the Finite Element method.
The energy potential of electrons at $V_{gs}=V_{th}$ in the normal gate transistor.¹

**The corresponding discussions in the revised manuscript:**

“The threshold voltage (V_t) decreased with V_{ds} increased because the stronger drain electric fields
penetrated into the channel region and thinned the barrier, resulting in compromising the gate control
capability, which was named by the drain-induced barrier lowering (DIBL) effect⁴⁶.” (Page.12, line 6)

**Reference:**

1. Liu, Y., Wang, P., Wang, Y., Huang, Y. & Duan, X. Suppressed threshold voltage roll-off and ambipolar
transport in multilayer transition metal dichalcogenide feed-back gate transistors. *Nano Res.* **13**, 1943–
1947 (2020).

**Comment 5:**

“The references in Figure 4 should be labeled at corresponding positions in Figure 4i.”

**Response:**

We thank the reviewer for the valuable suggestions. We have labelled the reference
numbers at corresponding positions as shown in **Figure 4i** in the revised manuscript.

**Figure 4i.** Value comparison of ideality factor n and off-state current of the MoS₂^{7, 20, 40-44} and WSe₂^{14, 45-}
⁴⁹ SJFET in previous reports.

**The corresponding discussions in the revised manuscript:**

**“Figure 4. i)** Value comparison of ideality factor n and off-state current of the MoS₂ and WSe₂
**SJFET in previous reports.”** (Page.11, line 13)

**Response to Reviewer 2#**

**General comments:**

*“In the manuscript, the authors reported the reconfigurable asymmetric Schottky junction FETs*
*using an ambipolar WSe₂-light photodetector. To achieve the ambipolar-to-unipolar polarity*
*conversion while suppressing the leakage current, the authors employed the electrostatically*
*tunable graphene and electric-field-shielding WTe₂ as top and bottom contacts to the WSe₂*
*channel with the laterally contacted FET geometry, respectively. The authors have also*
*performed intensive electrical and optoelectrical characterization for the suggested device*
*structure. However, despite the demonstration of high performance for rectification ratio and*
*off-state current than previous reports, I could not find any other remarkable result or*
*significant advances or structural novelty in this work, considering that many TMD-*
*heterojunction-based tunable rectifiers or photodiodes with similar device geometries have*
*been reported.”*

**Response:**

We thank the reviewer for the valuable comments and suggestions. In the revised
manuscript, we added more experiments, simulations, and discussions to provide the advance
of the *van der Waals* (vdW) offset metal-semiconductor-metal (MSM) geometry based on the
multilayer graphene (MGr) and semimetal WTe₂, **especially compared to the bulk metal**
**contact**. The runtime reconfigurable field-effect transistor (FET) originated from the ambipolar
transport behavior of 2D semiconductors¹. It usually suffered from the narrow switching-off
gate-voltage V_g range because of the continuous transition between electron-conduction and
hole-conduction states. That resulted in a challenge to realize the OFF state for logic functions
23². Besides, for the SJFET, the high symmetry of the transfer characteristics in p- and n-
24 configuration is also desired to ensure switching delay indifference³. This work provided an
25 alternative contact strategy to solve these problems. Compared to the barrier FETs in previous
reports^{4, 5}, our device showed a high symmetry of the transfer characteristics in p- and n-
configuration (both n- and p-type on-state $I_{ds} > 10^{-2} \mu A/\mu m$), which is desired to ensure
switching delay indifference. Besides, our device showed a pure n- or p-type behavior making
SJFET be switched off properly and showing a lower leakage current (high I_{ds} on/off ratio at V_g
$= \pm 50 V > 10^6$), which enabled the larger V_g switching range for dynamic configuration devices
with the sub picowatt static power consumption. Simultaneously realizing these characters was
not reported in previous works. **We have responded point by point to the reviewers' concerns,**
**added revisions, and highlighted them in the manuscript,** and hope that the revised manuscript
will address all your concerns.

**The main responses are summarized as follows:**

- ➤ We added more content to explain the difficulty of combining the reconfigurability and the
low leakage current in the SJFET. We also added more content to explain the working

- mechanism and emphasize the advantages of our device.
- ➤ We fabricated other WSe₂ FETs with different metal contacts to show that the MGr as the
 - top contact, possessed high and balanced carrier (electron and hole) injection efficiency.
 - ➤ We used Kelvin probe force microscopy, Raman spectral, and electrical measurement to
 - prove the semimetal WTe₂ bottom contact was important in SJFET due to its weaker
 - interlayer interaction to WSe₂, especially, compared to that of bulk metal such as Au.
 - ➤ We analyzed the current flow paths in the *vdW* contact region and explained the difference
 - between top and bottom contact geometry based on the transmission line model and the
 - COMSOL simulation.
 - ➤ We added the four-wire measurement to fairly compare the conductance of WTe₂ with
 - varying thicknesses.
 - ➤ We added more discussion on the effective field-effect mobility of SJFET and specified the
 - samples used in photo response measurement to make the demonstrations more concrete.

**Reference:**

- 1. Hu, W. *et al.* Ambipolar 2D Semiconductors and Emerging Device Applications. *Small Methods* **5**,
- 2000837 (2021).
- 2. Wu, P., Reis, D., Hu, X. S. &Appenzeller, J. Two-dimensional transistors with reconfigurable
- polarities for secure circuits. *Nat. Electron.* **4**, 45–53 (2021).
- 3. Fei, W., Trommer, J., Lemme, M. C., Mikolajick, T. &Heinzig, A. Emerging reconfigurable electronic
- devices based on two-dimensional materials: A review. *InfoMat* **4**, e12355 (2022).
- 4. Du, J. *et al.* Gate-Controlled Polarity-Reversible Photodiodes with Ambipolar 2D Semiconductors.
- *Adv. Funct. Mater.* **31**, 2007559 (2021).
- 5. Seo, S.-Y. *et al.* Reconfigurable photo-induced doping of two-dimensional van der Waals
- semiconductors using different photon energies. *Nat. Electron.* **4**, 38–44 (2021).

**Comment 1:**

*“The authors demonstrated reconfigurable WSe₂ SJFET with effectively suppressed leakage*

*current. However, I am not sure it presents methodological advances considering the previous*

*results. The previous method also shows a "shielding effect" via bottom vdW contact,*

*demonstrating a reconfigurable rectifier with low off-state leakage current $10^{-14} \sim 10^{-12}$ A at V_{ds}*

*= 1 V and gate-tunable photovoltaic property with V_{oc} of ~ 0.8 V. (Nano Energy 49 103-108 '18,*

*J. Mater. Chem. C, 6(43), 11673-11678 '18).”*

**Response:**

Thanks for the reviewer’s valuable comments. We added some comparisons to previous

reports to emphasize the methodological advances of our contact strategy based on *vdW*

semimetals. In previous reports, it was hard to achieve reconfigurable polarity and high

performance, e.g., the low leakage, simultaneously, which was relieved in our contact strategy.
 Generally, the polarities of 2D semiconductors were usually determined by the charge transfer
 doping from their dielectric environment or carrier injection control through junction interfaces.
 For the charge transfer doping, for example, although the in-plane WSe₂ P-N junction FET
 showed a large open-circuit voltage V_{oc} of ~ 0.8 V and the relatively low leakage currents,
 however, the direction of rectification could not be reversed due to the fixed local chemical
 doping as shown in **Figures R1a-c** (*J. Mater. Chem. C*, 6(43), 11673-11678 '18)¹. For the
 carrier injection controlling in the junction, the WSe₂/GeSe junction FET with gate-reversible
 rectification (*Nano Energy* 49 103-108 '18)². The junction FET showed the minimum leakage
 current of $\sim 10^{-13}$ A, however, the ambipolar transport behaviors were not suppressed as shown
 in **Figures R1d and 1e**. Besides, limited by the semiconductor nature of the GeSe and the band-
 alignment-evolutions, the on-state current density of I_{ds} was low and the p-branch (for example,
 at $V_{ds} = -1$ V) was hard to exhibit good performance matched to the strong n-branch as shown in
 **Figure R1f**.
 In this work, based on modulating the carrier injection through the contact barriers, we achieved
 the ambipolarity to reconfigurable-unipolarity transition, by appropriately optimizing contact
 materials and structures, meanwhile keeping a low leakage current as shown in **Figures R1g-**
 **h**, which is beneficial to switch off the SJFET more properly.

 **Figure R1. a)** Schematic of WSe₂ junction FET doped by the surface absorption. **b)** The I_d - V_d curve
 of the WSe₂ junction. **c)** V_{oc} and I_{sc} versus V_G curves. The sign of V_{oc} and I_{sc} were not switchable.¹
 **d)** Schematic of WSe₂/GeSe junction FET. **e)** The I_{ds} - V_{ds} curve of the WSe₂/GeSe junction FET. **f)**

The transport curves of the WSe₂/GeSe junction FET.² The n- and p-transfer curves were not
symmetric. **g**) Schematic of asymmetric contacted WSe₂ SJFET. **h**) Transfer curves of the device at
$V_{ds} > 0$ showed n-type polarity. **i**) Transfer curves of the device at $V_{ds} < 0$ showed p-type polarity. The
red dash lines indicated V_t .

**The corresponding discussions in the revised manuscript:**

“The reconfigurable rectifying operation was based on unpinned energy band...only holes were
allowed to be injected from the MGr side when $V_g < 0$.” (Page.13, line 27)

**References:**

- 1. Yang, Y., Huo, N. & Li, J. Gate-tunable and high optoelectronic performance in multilayer WSe₂ P–N
diode. *J. Mater. Chem. C* **6**, 11673–11678 (2018).
2. Yang, Z. *et al.* WSe₂/GeSe heterojunction photodiode with giant gate tunability. *Nano Energy* **49**, 103–
108 (2018).

**Comment 2:**

“In the suggested device structure, what are the advantages of using the WTe₂ contact instead
of metal electrodes? Schottky contacted metals can shield the gate electric field more effectively,
and the Fermi-level depinning is also achieved by vdW transferred metal contacts, according
to many recent reports. Additionally, it makes device fabrication much easier.”

**Response:**

Thanks for the reviewer’s valuable questions. Although the transferred metal was reported
to reduce the Fermi level pinning induced by the defect-induced-gap-state (DIGS) by avoiding
the defects generated in the metal deposition process¹. However, the 2D semiconductor still
suffered from the strong perturbation of extended wavefunction from the metal, known as the
metal-induced-gap-states (MIGS)². Both DIGS and MIGS induced the unpredictable charge
transfer doping for 2D semiconductors and degraded the performance of the 2D devices. The
semimetal such as WTe₂ has near-zero DOS at the Fermi level to reduce MIGS². The Fermi
level of a semimetal WTe₂ is close to the Fermi level of the semiconductor, the charge transfer
between M/S interfaces is expected to be reduced.

The advances of WTe₂ contact were also discussed in detail based on electronic characters,
KPFM, and Raman spectra. The WSe₂ SJFET with Au-bottom-contact exhibited poor
reconfigurability and leakage repression capability compared to WTe₂-bottom-contacted
SJFET as shown in **Figure S16 (Figure R2)**, although the Au and WTe₂ possessed similar work
functions (**Figures 2f-g, Figures R3a-b**). The potential of WSe₂ on Au film was higher than
WSe₂ on WTe₂ flake in **Figure S12b-e (Figures R3d-f)**, indicating the Fermi level of WSe₂ was
downward shifted due to the strong charge (hole) transfer doping effect of Au film. Besides, the

intensity of the characteristic peak (E_{2g}) of WSe₂ on Au film was decreased compared to that of
 WSe₂ on SiO₂ substrate, while the Raman intensity of WSe₂ on WTe₂ only showed a less
 decrease compared to the reference as shown in **Figures S13a-b (Figures R4a-b)**. Especially,
 the A_{1g} mode is easy to be affected by the electrostatic environment changes^{3,4}. The full width
 at half maximum (FWHM) of A_{1g} peaks was enlarged as the surface potential of WSe₂ increased
 ⁵ (**Figure S13e-f, Figures R4e-f**), indicating the stronger charger transfer doping effect on Au
 film.

 **Figure R2. a)** Schematic band diagram of bottom Au-WSe₂ Schottky junction. Due to the gap-states
 induced Fermi-level-pinning (FLP) at the WSe₂/Au interface, the contact resistance was hard to predict.
 **b)** Transfer curves of bottom-Au-contacted WSe₂ FET. The transfer curves showed the p-type transport
 characteristics. **c)** Photograph of the bottom-Au/WSe₂/top-MGr FET. **d)** Transfer curves of bottom-Au-
 contacted WSe₂ FET. The shapes of transfer curves were tuned by the V_{ds} .

 **Figure R3. a)** Height image of the Au/WTe₂ junction. **b)** Potential difference between the Au/WTe₂
 junction. **c)** Height image of the WSe₂ on Au film and WSe₂ on SiO₂ substrate. **d)** Potential difference
 between the WSe₂ on Au film and WSe₂ on SiO₂ substrate. **e)** Height image of the WSe₂ on SiO₂ substrate
 and WSe₂ on WTe₂ flake. **f)** Height image WSe₂ on SiO₂ substrate and WSe₂ on WTe₂ flake.

**Figure R4.** a) Raman spectra of the WSe₂ on the Au film and the SiO₂ substrate. The inset shows the
 photograph of the Au/WSe₂ junction. b) Raman spectra of the WSe₂ on the WTe₂ flake and the SiO₂
 substrate. The inset shows the photograph of the WTe₂/WSe₂ junction. Scale bar: 10 μm. c)-d) Raman
 peak and corresponding Lorentz fitting of WSe₂ on the Au film, WTe₂ flake, and the SiO₂ substrate. e)-f)
 FWHM of E_{2g} and A_{1g} characteristic peaks of WSe₂ on the Au film, WTe₂ flake, and the SiO₂ substrate
 as the reference. The insets show the corresponding vibration modes of WSe₂. Blue spheres: Selenium
 atom atoms. Black spheres: W atoms.

**The corresponding discussions in the revised manuscript:**

“The results suggested that the WTe₂ bottom contacts exhibited a low carrier injection efficiency due to
 the self-shielding effect of bottom contact¹⁴, weak interfacial interaction, and vdWs-gap-induced
 tunneling barrier at the WTe₂/WSe₂ interface. ...” (Page.9, line 14)

“Beyond the appropriate WF, as a typical semimetal, the few-layered WTe₂ also showed a weak
 interfacial interaction with the orbital overlapping to WSe₂, ... indicating the stronger charge transfer
 doping effect on Au film^{36, 37}” (Page.10, line 19)

**References:**

1. Liu, L. *et al.* Transferred van der Waals metal electrodes for sub-1-nm MoS₂ vertical transistors. *Nat.*
 *Electron.* **4**, 342–347 (2021).
 2. Shen, P.-C. *et al.* Ultralow contact resistance between semimetal and monolayer semiconductors.
 *Nature* **593**, 211–217 (2021).
 3. Zhao, W. *et al.* Lattice dynamics in mono- and few-layer sheets of WS₂ and WSe₂. *Nanoscale* **5**,
 9677–9683 (2013).

- 4. Buscema, M., Steele, G. A., van der Zant, H. S. J. & Castellanos-Gomez, A. The effect of the substrate
on the Raman and photoluminescence emission of single-layer MoS₂. *Nano Res.* **7**, 561–571 (2014).
5. Banszerus, L. *et al.* Identifying suitable substrates for high-quality graphene-based heterostructures.
*2D Mater.* **4**, 25030 (2017).

**Comment 3:**

*“In Figure 3, the charge injection ability of the graphene bottom electrode was compared with*
*the bottom electrode WTe₂. But is there any reason why graphene was used as the top electrode*
*in the final device structure? In this case, carrier injection through top graphene contact might*
*not be efficient enough because channel WSe₂ multilayers may shield the portion of the electric*
*field.”*

**Response:**

Thanks for the reviewer’s valuable questions and comments. For the offset contact strategy,
the top contact should have balanced electron and hole injection capability to achieve run-time
reconfigurability. The appropriate work function of the MGr rendered high injection efficiency
for both electron and hole, but other metals only had a stronger hole injection capability (WSe₂
FET with Au, PdSe₂, and NbSe₂ contacts in **Figure S8 (Figures R5a-c)**. MGr-contacted WSe₂
FET showed a symmetric ambipolar characteristic meanwhile keeping the high on-state current
(**Figure 3a, Figures R5d**). The transfer curves of WTe₂ contacted FET was also symmetric, but
the on-state current was limited. Hence, we used the MGr as the top contact (**Figure 3b, Figures**
**R5e**).

As for the contact geometry design, compared to the bottom contact, the top contact showed a
higher carrier injection capability¹. In the 2D FETs with the conventional top contact geometry,
the metal-semiconductor interface can be regarded as a resistor network under the diffusive
approximation (**Figures 3e-j, Figures R6 a and e**), and R_C is calculated according to the
transmission line model,

$$R_C = \sqrt{\rho_{sc} r_c} \coth(L_c / \sqrt{\rho_{sc} r_c}) \quad (1)$$

where ρ_{sc} is the sheet resistance of the 2D semiconductor beneath the contact, r_c is the specific
resistivity of the metal-semiconductor interface and L_c is the contact length, respectively.
Benefiting the global bottom gate, the ρ_{sc} were decreased as the positive or negative gate
voltage V_g increased, which improved the on-state current of 2D FETs. However, for bottom
contact geometry, the ρ_{sc} was hardly tuned by the bottom gate, resulting in large contact
resistance and poor on-state current density. This phenomenon was verified by simulation using
the COMSOL Multiphysics package as shown in **Figure S10 (Figures R6 b-h)**, in which the carrier
density of WSe₂ under the top contact was modulated effectively by the V_g hence the ρ_{sc} was
reduced.

**Figure R5. Transfer curves of the top contacted WSe₂ FET.** a) Transfer curves of the Au-contacted
FET. b) Transfer curves of the PdSe₂-contacted FET. c) Transfer curves of the NbSe₂-contacted FET. d)
Transfer curves of the MGr-contacted FET. e) Transfer curves of the WTe₂-contacted FET.

**Figure R6. a)** Schematic of the bottom contacted WSe₂ and the network of contact resistances. **b)-c)**
Distribution of electron density through the WSe₂ with bottom WTe₂ contacts, at $V_g=0$ and 100 V
respectively. **d)** Schematic band diagram of bottom contact geometry. **e)** Schematic of the top contacted
WSe₂ and network of contact resistances. **f)-g)** Distribution of electron density through the WSe₂ with
top WTe₂ contacts, at $V_g=0$ and 100 V respectively. **d)** Schematic band diagram of top contact geometry.
The dashed line in the band diagram indicates the band evolution induced by the V_g . Scale bar: 5 nm.

**The corresponding discussions in the revised manuscript:**

“Compared to the FET with top WTe₂ contacts, the FET with MGr top electrodes showed higher and
symmetric transfer curves, indicating the MGr possessed high carrier injection efficiency for both
electrons and holes, ...but most of these contacts showed an asymmetric carrier injection capability and
resulted in a stronger p-branch in the I_{ds} - V_g curves as shown in **Figure S8.**” (Page.9, line 5)

“To explain the contact-geometry-induced repression of carrier injection... resulting in a large contact
resistance and poor on-state current density.” (Page.9, line 14)

**Reference:**

1. Kong, L. *et al.* Doping-free complementary WSe₂ circuit via van der Waals metal integration. *Nat.*
 *Commun.* **11**, 1866 (2020).

**Comment 4:**

“In Fig. 4, the authors demonstrated reconfigurable unipolar behavior in WSe₂ SJFETs with
 asymmetric MGr/WTe₂ contacts. In this device structure, the authors claimed that WTe₂ shields
 the gate electric field effectively so that carriers cannot overcome the SBH at the junction
 between WSe₂ and WTe₂. However, considering the bottom contact geometry, the WSe₂ channel
 material at the junction on the edge of WTe₂ is affected by the gate, and the transport behaviour
 is likely to differ.”

**Response:**

Thanks for the reviewer’s valuable questions and comments. As the reviewer commented, the
 edge of WTe₂ may also contribute to the carrier injection. However, due to the large contact
 resistance with weak gate independence, the edge contact region contributed very little current
 in this device.

For the bottom contact geometry shown in **Figure R7a**, because the bottom edge of the WTe₂
 flake only had a line contact with the WSe₂ channel, the carrier injection quantum efficiency
 was low. More importantly, limited by the stiffness of the material and dry-transfer process, it
 is hard to fabricate the SJFET with the channel fitting the vertical side wall of the bottom contact
 well as shown in **Figure R7b**. The air gap also contributed to the large edge contact resistance
 r_e meanwhile it was hard to be modulated by the bottom gate V_g .¹ This large edge contact
 resistance makes the WSe₂ channel at the junction on the edge of WTe₂ unaffected by the gate.

**Figure R7. a)** Schematic of bottom contact resistances network. **b)** Schematic of the WTe₂ bottom
 contacted WSe₂ with a small gap at the edge. **c)** Schematic band diagram of bottom contact geometry.

**The corresponding discussions in the revised manuscript:**

“Besides, the van der Waals gap between the channel and the vertical side wall of the bottom contact

also contributed to the large and nonadjustable contact resistance due to the enlarged interface
resistance r_c .³⁴ (Page.9, line 29)

**Reference:**

1. Zhang, G. *et al.* Reconfigurable Two-Dimensional Air-Gap Barristors. *ACS Nano* 17, 4564–4573
(2023).

**Comment 5:**

Some other technical comments:

- In Fig. 2 i-k, it needs to specify the contact geometry. For a fair comparison, 4-terminal
conductance should be measured and compared.

**Response:**

Thanks for the reviewer's valuable suggestion. We specified the contact geometry of the
WTe₂ device in the revised manuscript. All devices were top contacted by the transferred Au
film as shown in **Figure S5 (Figures R8)**. We also added the 4-terminal conductance
measurement to compare the conductance of the WTe₂ with varying thicknesses fairly. **Figures**
**2h-g (Figure R9)** show the calculated resistance comparison measured by the 2-terminal and
4-terminal methods, in which both two resistances increased with the decrease in thickness.

**Figure R8.** **a)** Schematic of two-terminal electrical measurement. **b)** Schematic of four-terminal
electrical measurement. The transferred Au film was used as the electrode. **c)** Schematic of circuits of
four-terminal resistance measurement. **d)-f)** Photograph of the measured WTe₂ sample with varying
thicknesses. Scale bar: 10 μm . **h)-j)** Height images of measured WTe₂ samples with varying
thicknesses used in the electrical measurement.

**Figure R9. a)** I - V curves of W(Mo)Te₂ with varying thicknesses measured by the 2-terminal method. **b)**

I - V curves of W(Mo)Te₂ with varying thicknesses measured by the 4-terminal method. **b)** 2-terminal and

4-terminal resistance of W(Mo)Te₂ with varying thicknesses. The inset shows the contact resistance

$R_{contact} = R_{2T} - R_{4T}$ versus the thicknesses of W(Mo)Te₂.

**The corresponding discussions in the revised manuscript:**

“**Figures 2h and 2i** show the I - V curves of W(Mo)Te₂ with varying thicknesses measured by using the

2-terminal and 4-terminal methods respectively.... between the transferred Au film and WTe₂ also

increased with the decreased thickness as shown in the inset of **Figure 2j**.” (Page.6, line 24)

**Comment 6:**

- *Field-effect mobility extraction may not be appropriate using equation (3) because the*

*Schottky barrier will also vary with gating.*

**Response:**

Thanks for the reviewer’s valuable suggestion. The effective field-effect mobility

$\mu_{eff} = L_{ch}g_m / (W_{ch}C_{ins}V_{ds})$ is used in this work. As the reviewer commented, the μ_{FET} possibly

underestimated or overestimated relative to the drift mobility of the channel material depending

on the details of gate capacitance, V_{ds} , V_{gs} , L_{ch} , and especially R_c (including carrier injection

efficiency) in devices with short channel lengths, in which μ_{eff} was strongly limited by contact

resistance^{1, 2}.

Hence, in this work, we named the mobility as effective field-effect mobility μ_{eff} to emphasize

that both calculated electron and hole channel mobility were limited by contact barrier, instead

of as an indicator of performance in reconfigurable FETs. By increasing the V_{ds} , the μ_{eff} of hole

and electron were increased correspondingly due to that the shape of the contact barrier was

thinned.

**The corresponding discussions in the revised manuscript:**

“**Figure 4d** shows that the effective μ_{FET} of electron for WSe₂ FET was almost twice as high as the

effective μ_{FET} -hole of the device with the MGr contacts and both deduced μ_{FET} of electron and hole were

strongly influenced by V_{ds} since the mobility in SJFET was limited by the contact barrier.” (Page.13,

line 1)

**References:**

1. Cheng, Z. *et al.* How to report and benchmark emerging field-effect transistors. *Nat. Electron.* **5**, 416–
423 (2022).

2. Das, S. *et al.* Transistors based on two-dimensional materials for future integrated circuits. *Nat.*
*Electron.* **4**, 786–799 (2021).

**Comment 7:**

- *Output characteristics of the device at variable voltage are investigated in Figure 4 g and*
*Figure 5 a, b. However, the gate-dependent rectification properties of the WSe₂ diode are*
*inconsistent with each other for the dotted line representing dark current in Figure 4 g and*
*Figure 5 a, b. As the author indicated the WTe₂ as a drain contact, it would be better to correct*
*the results in Figure 5 a, b.*

**Response:**

Thanks for the reviewer’s valuable reminder. We used two devices to characterize the gate-
tunable photo responses of the SJFET. The photos of the device are shown in **Figure S11a-b**
**(Figure R10)**. The data in **Figure 5d-i** are derived from sample 1[#] which was also used to
measure the corresponding data in **Figure 4**. The data in Fig. 5a-c are derived from sample 2[#].
In the revised manuscript, we specified the data sources to make the demonstrations more
concrete.

**Figure R10.** a) Photograph of the WSe₂ FET in offset contact geometry. b) Photograph of the WSe₂
FET in offset contact geometry. Scale bar: 10 μm.

**The corresponding discussions in the revised manuscript:**

“We used two devices to investigate the photo response and the photographs are shown in **Figures**
**S10a-b**. Only the data in **Figure 5a-c** were derived from sample 2[#].” (Page.14, line 15)

**Response to Reviewer 3#**

**General comments:**

*“In this manuscript, the authors designed an offset Schottky junction field-effect transistor with*
*asymmetric contact of semimetal graphene and WTe₂, where they are top and bottom contact,*
*respectively. They claimed that the design can suppress the off-state current and control the*
*polarity of the transistor. After carefully reading the manuscript, I think the manuscript can not*
*be accepted for publication at such a situation.”*

**Response:**

We thank the reviewer for the comments and valuable suggestions on our review. We
carefully rechecked the manuscript and graphs and revised some typos to improving the
readability according to the reviewer’s comments. The runtime reconfigurable polarity in
Schottky junction field-effect transistor (SJFET) originated from the ambipolar transport
behavior of 2D semiconductors. However, the ambipolar transistor can only be turned off
within a narrow gate-voltage range because of the continuous transition between electron
conduction and hole-conduction states, resulting in a challenge to realize the OFF state for logic
functions. Besides, for the SJFET, the high symmetry of the transfer characteristics in p- and n-
configuration is also desired to ensure switching delay indifference. To solve these problems,
we used *vdW* offset contact strategy to fabricate the SJFET which possessed the simple MSM
structure, better unipolarity (high I_{ds} on/off ratio at $V_g = \pm 50 \text{ V} > 10^6$), and more symmetric n-
and p-transfer properties (both n- and p-type on-state $I_{ds} > 10^{-2} \mu\text{A}/\mu\text{m}$).

In the revised manuscript, we added the experiments, schematics, and discussions to address
the Schottky junction field-effect transistor (SJFET) working principles and advantages with
*van der Waals (vdW)* offset contact geometry. Especially, we added the experiments and
discussion to show the origination of the polarity control (n-type or p-type) for multilayer
graphene (MGr)/WSe₂/WTe₂ SJFET. We also compared our SJFET to the devices with
traditional metal contact in previous reports, e.g. *Adv. Funct. Mater.* 31, 2007559, 2021.
Benefiting from the *vdW* offset contact of MGr and WTe₂, the SJFET showed better
ambipolarity-to-reconfigurable-unipolarity conversion and the more symmetric on-state
current density I_{ds} for n- and p-type transport. The n-branch on-state I_{ds} of our device was at the
same scale compared to the previous report and p-branch on-state I_{ds} was much higher than that
with the bulk metal contacts.

As for the detailed issues, we have responded point by point to the reviewers’ concerns, added
revisions, and highlighted them in the manuscript, and hope that the revised manuscript will
address all your concerns.

**The main responses are summarized as follows:**

- ➤ We redrew the figures and changed the figure format to avoid image blurring. We also
carefully rechecked the manuscript and revised the typos and spelling errors.

- ➤ We used current density at the same V_{ds} to evaluate and compare the conductance of the
W(Mo)Te₂ and the MGr.
- ➤ We fabricated the MGr/WSe₂/WTe₂ SJFET in both top contact and bottom contact geometry,
which indicated that both were not beneficial to simultaneously achieving the ambipolarity
to unipolarity conversion and a high I_{ds} on/off ratio.
- ➤ We removed part of the content of WTe₂ growth and characterization into the
supplementary file.
- ➤ We provided the photographs of our devices and added the detailed schematic band diagram
and barrier calculation to explain the working mechanism of the reconfigurable SJFET.
- ➤ We compared the on-state current density I_{ds} of our WSe₂ SJFET with that in the previous
reports. Our device showed similar n-branch I_{ds} and far higher p-branch on-state I_{ds}
compared to the bulk metal contacted FET (*Adv. Funct. Mater.* 31, 2007559, 2021).
- ➤ We explained asymmetric I_{ds} - V_{ds} curves in symmetric contacted FET, mainly induced by
the different contact lengths.

**Comment 1:**

*“Most of the pictures are blurred and difficult to see clearly. Maybe high-resolution pictures
are better.”*

**Response:**

Thanks for the reviewer’s valuable reminder. We revised the format of the pictures to show
the result clearly.

**Comment 2**

*“Lots of errors make the readability of the manuscript not good. What does the sentence (line
20-26) mean? Figure 1a: symmetric or asymmetric (line 134)? Figure S8: WSe₂ or MoS₂? Line
266: “However, the off-state current was suppressed to 10^{-10} $\mu A/cm^{-1}$ even at $V_{ds} = 1V$ and V_g
$= -60 V$ ”, and more.”*

**Response:**

Thanks for the reviewer’s valuable questions and reminder. We carefully rechecked the
contents in the manuscript, the figures, and the Supplementary document and revised the
corresponding contents according to the reviewer’s reminders.

**The corresponding discussions in the revised manuscript:**

*“Here, we fabricated the tungsten diselenide (WSe₂) Schottky junction field-effect-transistor (SJFET) in
the van der Waal offset contact geometry ... the off-state current of SJFET was repressed to 2×10^{-10} μA
36 μm^{-1} and the static leakage power consumption was suppressed to 10^{-5} nW under 1 V drain bias.” (Page.1,
line 20-26)*

“Meanwhile, the off-state current was suppressed to $\sim 10^{-10}$ $\mu\text{A}/\mu\text{m}^{-1}$ at $V_{\text{ds}} = 1\text{V}$ and $V_{\text{g}} = -60\text{V}$, hence, a
maximum on/off ratio of $>10^6$ was achieved.” (Page.12, line 7)

“Figure 1. a) Schematic of global gated Schottky junction FET with symmetric contacts.” (Page.5, line
19)

“The SJFET also worked as a gate-tunable Schottky rectifier with a near-unity ideality factor of ~ 1.0 and
a high rectifying ratio of 3×10^6 .” (Page.3, line 17)

“The bulk sample exhibited a constant source-drain current I_{ds} when the gate voltage V_{g} swept,
showing a weak p-type characteristic, especially at the low temperature.” (Page.7, line 10)

“Figure S11. Surface potential measurement of WSe₂ on the different substrates by Kelvin Probe
Force Microscopy.” (Supplementary Information, Page.14, line 2)

12 **Comment 3:**

“What is the unit of conductivity, mS or S/m? The authors claimed that WTe₂ possesses high
conductance and appropriate work function to enhance the shielding effect in comparison with
graphene. Can the authors give the conductivity of their graphene to support their conclusion?”

**Response:**

Thanks for the reviewer’s valuable reminders and suggestions. The unit conductivity here
should be S or mS. We also compared the conductance of MGr and WTe₂ with a similar
thickness (around 20 nm). The $I_{\text{d}}-V_{\text{d}}$ curves indicated that the WTe₂ showed a comparable but
slightly smaller conductance compared to the MGr. This emphasized that the WTe₂ as the
semimetal, possessed the high conductance among the transition metal dichalcogenide (TMD)
family. The WTe₂ flake showed a high current density ($\sim 8\text{ mA}/\mu\text{m}$) at $V_{\text{bias}} = 0.1\text{ V}$, which was
far higher than the on-state I_{ds} of the WSe₂ SJFET ($< 10^{-2}\text{ mA}/\mu\text{m}$). Hence conductance of the
WTe₂ was satisfied to work as the bottom contact of SFFET in this work.

**Figure R1. a)** Optical image of the MGr sample used in the conductance measurements. Scale bar: 10
27 μm. **b)** AFM height image of the MGr. **c)** Current density I_{ds} versus V_{ds} of the MGr. The current density

$I_{ds} = I_{measured}/W$, and W is the width of the sample. **d)** Optical image of the MGr sample used in the
conductance measurements. Scale bar: 10 μm . **e)** AFM height image of the WTe₂ measured by the AFM.
**f)** Current density I_{ds} versus V_{ds} of the WTe₂. The current density $I_{ds} = I_{measured}/W$, and W is the width of
the sample.

**The corresponding discussions in the revised manuscript:**

“WTe₂ possessing an appropriate work function and weak interlayer interaction with WSe₂ is
expected to enlarge the tunneling width of SB at WTe₂/WSe₂ interface.” (Page.5, line 11)

“**Figures 2h and 2i** show the *I-V* curves of W(Mo)Te₂ with varying thicknesses measured by using
the 2-terminal and 4-terminal methods respectively.... between the transferred Au film and WTe₂
also increased with the decreased thickness as shown in the inset of **Figure 2j**.” (Page.6, line 24)

**Reference**

1. Mleczko, M. J. *et al.* High Current Density and Low Thermal Conductivity of Atomically Thin
Semimetallic WTe₂. *ACS Nano* **10**, 7507–7514 (2016).

**Comment 4:**

“Following comment 3, maybe the polarity control is mostly from the asymmetric work function
of graphene and WTe₂ like asymmetric metal electrodes? The authors should compare their
offset device with devices of asymmetric contact, both bottom and top.”

**Response:**

Thanks for the reviewer’s valuable question and suggestion. According to the reviewer’s
suggestion, we added the electrical measurements of the asymmetric MGr-WSe₂-WTe₂ SJFET
with top contact and bottom contact geometry (**Figure S15**, **Figure R2**). **Figure S15c** (**Figure**
**R2c**) shows the transfer curves of the SJFET in top contact geometry, in which the polarities
were hard to be controlled by the V_{ds} due to the high carrier injection efficiency of the top
contacts. **Figure S15d** (**Figure R2d**) shows the transfer curves of the SJFET in bottom contact
geometry, in which the polarity was controlled by the V_{ds} , however, the ambipolarity is hard to
be suppressed without the offset contact geometry.

Hence, although the MGr/WSe₂/WTe₂ SJFET in all-top and all-bottom contact showed weak
V_{ds} -dependent ambipolarity transport properties, all-top-contact or bottom-contact geometry
was not benefitted to simultaneously achieve the ambipolarity to unipolarity conversion (low
leakage current) and a high I_{ds} on/off ratio.

**Figure R2.** a) Optical image of the MGr/WSe₂/exfoliated-WTe₂ SJFET in top contact geometry. b)
Optical image of the MGr/WSe₂/WTe₂ SJFET in bottom contact geometry. c) Schematic of the
MGr/WSe₂/exfoliated-WTe₂ JFET in top contact geometry and the transfer curves at varying V_{ds} . d)
Schematic of the MGr/WSe₂/ WTe₂ SJFET in bottom contact geometry and the transfer curves at varying
V_{ds} .

**The corresponding discussions in the revised manuscript:**

“To verify the necessity of the offset contact geometry, we also measured the asymmetric FET in the top
contact and bottom geometry, both of which were not benefitted to simultaneously achieve the
ambipolarity to unipolarity conversion (low leakage current) and a high I_{ds} on/off ratio (Figure S15) ...
The asymmetric MGr/Au contacted FET showed poor reconfigurability which verified the WTe₂ indeed
played an important role in the polarity control.” (Page.12, line 19)

**Comment 5:**

“Maybe moving the part of WTe₂ growth to the supplementary materials is better.”

**Response:**

Thanks for the reviewer’s valuable and constructive suggestion. We moved a large part of
the content of WTe₂ growth into the revised Supplementary file.

**The corresponding discussions in the revised manuscript:**

“We used high-resolution transmission electron microscopy (HRTEM) and Raman spectra to probe the
quality of WTe₂... which means the MoTe₂/ WTe₂ heterojunctions can work as ultrathin van der Waals
contacts to build 2D Schottky junctions.” (Supplementary Information, Page.2, line 16)

**Comment 6:**

“The authors should provide pictures of their devices, which may help readers well understand
their work.”

**Response:**

Thanks for the reviewer’s valuable and constructive suggestion. According to the
reviewer’s suggestions, we added the photograph of our device (Figures S11a, Figure R3a)

and redraw the schematic of the device in the revised manuscript to show our work more clearly.
 **Figure S11c (Figures R3b-d)** shows the height image of SJFET to introduce the detailed
 information on our device, e.g. thickness.

 **Figure R3. a)** Optical image of the MGr/WS₂/ WTe₂ SJFET in top contact geometry. **b)** Schematic of
 the MGr/WS₂/WTe₂ SJFET in offset contact geometry. **c)** AFM height image of MGr/WS₂/WTe₂
 SJFET. Scale bar 2 μm. **b)** Height profile of the MGr, WS₂, and WTe₂ along the dash lines in **Figure**
 **R3c.**

**The corresponding discussions in the revised manuscript:**

**“Figure 4. Reconfigurable WSe₂ SJFET with asymmetric MGr/WTe₂ contacts. a)** Schematic of
 asymmetric contacted WSe₂ SJFET.” (Page.11, line 6)

**“Figure S11. Height and potential measurement of the WSe₂/WTe₂ and MGr/WTe₂. a)-b)**
 Photograph of the WSe₂ FET in offset contact geometry. Scale bar: 10 μm.” (Supplementary Information
 Page.13, line 1)

**Comment 7:**

*“The authors should give a more detailed explanation of the working principle of the offset*
 *device, like combing schematic pictures, which should help readers to well understand the*
 *device.”*

**Response:**

Thanks for the reviewer’s valuable suggestion. We added a detailed explanation of the
 working principle of the MGr/WS₂/WTe₂ SJFET in offset contact geometry. **Figure S18**
 **(Figure R4)** shows the schematic band diagrams of the SJFET. When $V_g > 0$ at $V_{ds} = -1$ V, the
 gate-electric field induced strong electron accumulation and reduced the surface potential ψ_s of the
 MGr-contacted WSe₂. Hence the SB width was thinned, which promoted the electron injection from
 MGr through the tunneling (**Figure S18b, Figure R4b**). In contrast, when $V_g < 0$, the width of the
 barrier at the WTe₂/WSe₂ interface remained almost constant due to the shielding effect as discussed
 in the revised manuscript Part I and III, which reduced the off-state hole current leakage (**Figure**

**S18c, Figure R4c).** Reversed carrier injection process happened at $V_{ds} < 0$ (**Figures S18d-f, Figure**
 **R4d-f**), only holes were allowed to be injected from the MGr side when $V_g < 0$.

 **Figure R4. a)** Schematic of the n-type SJFET ($V_{ds} = -1$ V). **b)** Schematic band diagram of the n-type
 SJFET at $V_g = 40$ V (on-state). **c)** Schematic band diagram of the n-type SJFET at $V_g = -40$ V (off-state).
 **d)** Schematic of the p-type SJFET ($V_{ds} = 1$ V). **e)** Schematic band diagram of the p-type SJFET at $V_g =$
 40 V (off-state). **f)** Schematic band diagram of the p-type SJFET at $V_g = -40$ V (on-state).

**The corresponding discussions in the revised manuscript:**

“The reconfigurable rectifying operation was based on unpinned energy band at the MGr/WSe₂ interface
 and the strong carrier-injection suppression capability of WTe₂.... only holes were allowed to be injected
 from the MGr side when $V_g < 0$.” (Page.13, line 27)

**“Figure S18. Operating mechanism of reconfigurable MGr-WSe₂-WTe₂ SJFET in offset contact**
 **geometry.”** (Supplementary Information Page.19, line 1)

**Comment 8:**

“The authors claimed that the structure could suppress the off-state current. From the
 manuscript, it can be seen that the on/off ratio is not high. Does the structure also suppress the
 on-state current? This is not good for device applications. How does it compare with metal
 electrodes like *Adv. Funct. Mater.* 31, 2007559 (2021)?”

**Response:**

Thanks for the reviewer’s valuable question and suggestion. The on-state I_{ds} of the SJFET
 was limited by the large channel length (>5 μm) of the WSe₂. Compared the conventional WSe₂
 FET in all-top contact geometry, The on-state I_{ds} of SJFET in offset contact geometries did not
 apparently degrade. (**Figure S14, Figures R5a-c and R6**).

We also compared the on-state I_{ds} with the result in previous reports. Our device still showed
 comparable on-state I_{ds} to the conventional WSe₂ ambipolar transistor ¹ (**Figures R5d-e**).
 Especially, compared to the SJFET with metal contacts (*Adv. Funct. Mater.* 31, 2007559, 2021),
 our device not only showed the better unipolar transfer characteristic but also the higher on-

state current density at $V_{ds} = \pm 1V$, especially for hole current density (p-type) in **Figure R5f** ².

**Figure R5. a)** Transfer curves of the MGr-contacted FET at varying V_{ds} . **b)** Transfer curves of the WTe₂-
contacted FET at varying V_{ds} . **c)** Transfer curves of the SJFET in the offset contact geometry at varying
V_{ds} . **d)** Transfer curves of single-layer WSe₂ FET with deposited metal electrodes. **e)** Transfer curves of
multilayered WSe₂ FET with deposited metal electrodes. **f)** Transfer curves of Au-WSe₂-In SJFET in
offset contact.

**Figure R6. Comparison of the on-state I_{ds} among the symmetric and asymmetric contacted WSe₂**
**FET.**

**The corresponding discussions in the revised manuscript:**

“Meanwhile the on-state I_{ds} of SJFET in offset geometry were not decreased compared with that of
WSe₂ FET with the symmetric top contact geometry as shown in **Figure S14.**” (Page.12, line 7)

“**Figure S14. Comparison of the on-state I_{ds} among the symmetric and asymmetric contacted WSe₂**
**FET.**” (Supplementary Information Page.11, line 1)

**References:**

1. Wang, Z. *et al.* The ambipolar transport behavior of WSe₂ transistors and its analogue circuits. *NPG*

*Asia Mater.* **10**, 703–712 (2018).

2. Du, J. *et al.* Gate-Controlled Polarity-Reversible Photodiodes with Ambipolar 2D Semiconductors.

*Adv. Funct. Mater.* **31**, 2007559 (2021).

**Comment 9:**

“Figure 3 is for symmetric contact devices. Theoretically, the transport should be independent
on the direction of V_{ds} . In Figure 3c and 3d, the results exhibit that they are dependent. Is this
from the difference of the source and drain? I mean maybe the contact area, thickness and
others are different and affect the device performance.”

**Response:**

Thanks for the reviewer’s comments and reminder. As the reviewer commented, the
transfer curves should be identical at positive and negative V_{ds} for a perfect symmetric contacted
device. For our device in **Figure S9 (Figure R7)** although the contact materials were identical,
however, the contact lengths were different (marked in red dashed circles), which induced that
performance of our device deviated from that of the ideal device. In fact, as the reviewer
commented, apart from the contact metal, the properties of the contact region of 2D
semiconductors including the shape, area, and thickness can influence the transport property by
affecting the contact barrier^{1, 2}. But compared to contact materials, the asymmetric geometry
of WSe₂ only had a weak influence on the polarity control.

**Figure R7. a)** Photograph of the WSe₂ FET in bottom contact geometry. Scale bar: 10 μm . **b)** I_{ds} - V_{ds}
curves of MGr-contacted WSe₂ FET at V_{gs} . **b)** I_{ds} - V_{ds} curves of MGr-contacted WSe₂ FET at V_{gs} .

**The corresponding discussions in the revised manuscript:**

“The asymmetric I_{ds} - V_{ds} curves were induced by the different contact lengths at the source and
drain side.” (Supplementary Information Page.13, line 5)

**References:**

1. Zhou, C. *et al.* Self-Driven Metal–Semiconductor–Metal WSe₂ Photodetector with Asymmetric
Contact Geometries. *Adv. Funct. Mater.* **28**, 1802954 (2018).

2. Cheng, Z. *et al.* Distinct contact scaling effects in MoS₂ Transistors revealed with asymmetrical contact
measurements. *Adv. Mater.* 2210916 (2023).

REVIEWERS' COMMENTS

Reviewer #1 (Remarks to the Author):

My questions and concerns have been addressed by the authors. The manuscript has been satisfactorily revised. Based on the major improvement made, I would like to pass it for acceptance.

Reviewer #2 (Remarks to the Author):

In the revised manuscript, I think that the authors have made efforts to address most of our concerns in their responses. If they can provide satisfactory answers to the remaining technical comments, I recommend the revised manuscript for publication in Nature Communications.

1. To clearly visualize the device operation process, the authors should mark which one is Drain or Source in the schematics of the offset contact and WTe₂ contact in Figure 1. For the bottom contacted magenta & purple material and the top contacted magenta material, it is necessary to indicate which one represents the Drain and Source.

2. Regarding the question about the advantages of top WTe₂ compared to bottom metal, the previous response incorrectly compared bottom metal and top graphene instead of comparing metal and WTe₂. The authors should compare the advantages of using WTe₂ as the top contact compared to using metal as the bottom contact to address the question accurately.

Reviewer #3 (Remarks to the Author):

After reviewing the revised manuscript, I think it may be acceptable for publication now.

Response to Reviewers' Comments

Response to Reviewer 1#

General comments:

“My questions and concerns have been addressed by the authors. The manuscript has been satisfactorily revised. Based on the major improvement made, I would like to pass it for acceptance.”

Response:

We thank the reviewer for the valuable comment and appreciation. We appreciate the reviewer's time and suggestion for our work again.

Response to Reviewer 2#

General comments:

“In the revised manuscript, I think that the authors have made efforts to address most of our concerns in their responses. If they can provide satisfactory answers to the remaining technical comments, I recommend the revised manuscript for publication in Nature Communications.”

Response:

We thank the reviewer for the valuable comments and suggestions. In the revised manuscript, we added corresponding labels to indicate the drain and source in the schematics. We also added discussions about the comparison between the WTe₂-top-contacted SJFET and Au-bottom-contacted SJFET to further address the role of WTe₂. **We have responded point by point to the reviewers' concerns, added revisions to the manuscript,** and hope that the revised manuscript will address all your concerns.

The main responses are summarized as follows:

- We revised Figure 1 to indicate the drain and source in the schematics of offset contact geometry.
- We compared the electrical performance of SJFET with WTe₂ as the top contact and Au film as the bottom contact to address the role of WTe₂ accurately.

Comment 1:

“To clearly visualize the device operation process, the authors should mark which one is Drain or Source in the schematics of the offset contact and WTe₂ contact in Figure 1. For the bottom contacted magenta & purple material and the top contacted magenta material, it is necessary to indicate which one represents the Drain and Source.”

Response:

Thanks for the reviewer's valuable suggestion. We revised Figure 1 to address your concern by indicating the drain and source in the schematics of offset contact geometry. For

offset-contacted FETs in this work, the top contacts represent the drain, and the bottom contacts represent the source.

Figure 1. The transition from ambipolarity to unipolarity behavior of the Schottky junction field-effect transistor (SJFET).

The corresponding discussions in the revised manuscript:

“Figure 1. The transition from ambipolarity to unipolarity behavior of the Schottky junction field-effect transistor (SJFET).” (Page.20, line 2)

Comment 2:

“Regarding the question about the advantages of top WTe₂ compared to bottom metal, the previous response incorrectly compared bottom metal and top graphene instead of comparing metal and WTe₂. The authors should compare the advantages of using WTe₂ as the top contact compared to using metal as the bottom contact to address the question accurately.”

Response:

Thanks for the reviewer’s valuable suggestions. The top WTe₂ electrode enabled WSe₂ FETs to show a more symmetric transfer curve compared with the bottom bulk metal electrode, e.g., Au film. To address the advantage of the top WTe₂ contact, we compared the performance of the WTe₂-top-contacted SJFET and Au-bottom-contacted SJFET, the results showed SJFET using WTe₂ as the top contact shows the balanced ambipolar transfer characteristics as shown in Figure R1a and 1b (Figure S16b and 3b), which indicated the work function of WTe₂ was nearly aligned to the Fermi level of WSe₂. Furthermore, Figure R1c (Figure S16f) shows the V_{ds}-dependent transfer curves of an asymmetrically contacted WSe₂ FET with top WTe₂ and bottom Au contacts. Due to the Fermi level pinning at the interface of bulk metal and WSe₂, the ambipolar transfer characteristics were significantly suppressed.

Figure R1. Transfer curves of WSe₂ FET with the Au and WTe₂ contacts. a) Transfer curves of Au bottom-contacted FET. b) Transfer curves of WTe₂ top-contacted FET. c) Transfer curves of asymmetrically contacted FET.

The corresponding discussions in the revised manuscript:

“Supplementary Figure 16. Electrical measurement of bottom-Au-contacted WSe₂ FET. e) Optical image of the bottom-Au/WSe₂/top-exfoliated-WTe₂ FET. f) Transfer curves of bottom-Au/WSe₂/top-exfoliated-WTe₂ FET.” (*Supplementary Information, Page.20, line 2*)

Response to Reviewer 3#

General comments:

“After reviewing the revised manuscript, I think it may be acceptable for publication now.”

Response:

We thank the reviewer for the valuable comment and appreciation.